# Capacity and Bias of
# Learned Geometric Embeddings for Directed Graphs

**Michael Boratko**[*,1]**, Dongxu Zhang**[*,1]**, Nicholas Monath**[†,1]**, Luke Vilnis**[†,1]**,
Kenneth L. Clarkson**[2]**, Andrew McCallum**[1]
[1] University of Massachusetts Amherst
[2] IBM Research
{mboratko, dongxuzhang, nmonath, luke, mccallum}@cs.umass.edu
klclarks@us.ibm.com

## Abstract

A wide variety of machine learning tasks such as knowledge base completion, ontology alignment, and multi-label classification can benefit from incorporating into learning differentiable representations of graphs or taxonomies. While vectors in Euclidean space can theoretically represent any graph, much recent work shows that alternatives such as complex, hyperbolic, order, or box embeddings have geometric properties better suited to modeling real-world graphs. Experimentally these gains are seen only in lower dimensions, however, with performance benefits diminishing in higher dimensions. In this work, we introduce a novel variant of box embeddings that uses a learned smoothing parameter to achieve better representational capacity than vector models in low dimensions, while also avoiding performance saturation common to other geometric models in high dimensions. Further, we present theoretical results that prove box embeddings can represent any DAG. We perform rigorous empirical evaluations of vector, hyperbolic, and region-based geometric representations on several families of synthetic and real-world directed graphs. Analysis of these results exposes correlations between different families of graphs, graph characteristics, model size, and embedding geometries, providing useful insights into inductive biases of various differentiable graph representations.

## 1 Introduction

Embedding-based methods are a cornerstone of modern approaches to modeling graph data. Geometric embeddings, which represent elements using objects with richer geometric structure (such as boxes [60], cones [27, 58], discs [55], densities [2, 59], and elements of hyperbolic space [39, 40]) have recently been proposed as a powerful alternative to standard vector embedding models for representing asymmetric relationships. Modeling such asymmetric relationships is central to discovering embedded representations of directed graphs, such as DAG-structured taxonomies, where geometric embeddings have been shown empirically to be more effective [39, 58, 60].

The success of geometric embeddings largely rests on exploiting inherent biases intrinsic to the particular geometry chosen. The negative curvature of hyperbolic space, for example, provides an inductive bias which is believed to better model the exponential increase of nodes in trees [39, 61, 62]. Region-based representations, on the other hand, model directional edges via containment relationships [58, 60]. As containment is, itself, transitive, these models would seemingly have a natural bias toward representing transitive relationships. While previous work has empirically tested

---

[*] Equal Contribution. [†] Now at Google.

35th Conference on Neural Information Processing Systems (NeurIPS 2021).

the generalization capability of these region-based geometric representations on directed graphs with missing edges [17, 27, 30, 58, 60] an understanding of their representational capability, both theoretically and experimentally, is lacking.

There are several notable open questions regarding geometric embeddings. First, previous work has observed that the performance benefits of geometric embeddings are most strongly observed in low dimensions, and are often diminished or negated by vector models in higher dimensions, raising the question of *whether there exists a geometric embedding model which can utilize higher dimensions more effectively*. Second, previous work has provided theoretical results for vector [4, 15] and hyperbolic [48, 49] models, but has not addressed *what region-based embeddings, such as order and box embeddings, can represent, theoretically*. Third, there is not an agreed upon convention for *how to empirically analyze the bias and representational capacity of models without confounding factors such as optimization, hyperparameters, and generalization performance.*

In this paper, we address each of these open questions. We introduce a novel box embedding model, extending the recently proposed Gumbel-box process [17] to learn from data a parametric model representing a continuous relaxation of box intersections. In this way, we can effectively rely on the structure of the graph to guide the fitting of the embedding model. Our model can trade-off between smoother, vector-like settings and more discrete, region-like settings, providing better performance across both lower and higher dimensionalities in a single model. We provide theoretical analysis of this approach as well as region-based approaches more generally. We lastly design a rigorous evaluation framework, which uses Bayesian hyperparameter optimization to fairly compare representational capacity independently from optimization details of each approach. This empirical framework allows us to characterize properties of graphs for which each method performs well.

In summary, our contributions are as follows:

**Methodological Contributions (§3)** We introduce a novel continuous relaxation of probabilistic box embeddings, where each node has an associated trained variance or temperature.

**Theoretical Contributions (§4)**: We prove

- Order Embeddings may produce cycles, even when pruning reverse edges (Proposition 2)
- Probabilistic Order Embeddings (or Order Embeddings under $\ell^1$ energy) always produce acyclic graphs when pruning reverse edges (Theorem 1)

While it is known [60] that box embeddings also produce acyclic graphs when reverse edges are removed, we significantly strengthen this by proving

- *Any* DAG can be represented by Probabilistic Box Embeddings in $\mathcal{O}((\Delta + 2) \log |V|)$ dimensions, where $\Delta$ is the maximum node degree, and $|V|$ the number of nodes (Theorem 2)
- This can be accomplished using $\mathcal{O}(D(\Delta + 2) \log^2 |V|)$ bits of precision, where $D$ is the max depth (Proposition 3)

**Empirical Highlights (§6)**: We conduct a suite of experiments on synthetic and real-world graphs.[1] We compare various vector, hyperbolic, cone, and box representations over a range of dimensions, using extensive Bayesian hyperparameter tuning, and find that

- Geometric embedding models perform better in low dimensions
- Our model outperforms all other geometric embeddings in high dimensions
- Our model always outperforms the state-of-the-art box embeddings when using equal dimensions
- Hyperbolic models do particularly well on trees, but struggle with transitivity
- Region-based embeddings perform particularly well on transitively-closed graphs

## 2   Background: Embedding Directed Graphs

Given a simple[2] directed graph $G$ with nodes and edges $(V, E)$ we seek to represent the adjacency matrix, that is, a function $f : V \times V \to \{0, 1\}$ where $f(i, j) = 1$ if $(i, j) \in E$ and 0 otherwise. We parameterize the nodes, defining a function $\phi : V \to \mathbb{R}^d$, and define an energy function $\mathrm{E} : \mathbb{R}^d \times \mathbb{R}^d \to \mathbb{R}_+$, which models the negative log probability of edge existence. We can view

---

[1]Our code and data are available at `https://github.com/iesl/geometric_graph_embedding`.

[2]A *simple* directed graph has no multiple edges or self-loops, i.e. the adjacency matrix contains only 0s and 1s, with 0s on the diagonal. We remove self-loops from the learned representations.

the probabilities provided as a weighted graph, however in practice it is often necessary to make a hard decision on edge existence, which is done by choosing a (global) threshold $\tau$ and binarizing the output, yielding an adjacency matrix $A_{i,j} = \mathrm{E}(i,j) < \tau$.

## 2.1 Vectors

A straightforward approach to this problem would be to parameterize each node using a vector, however typical score functions for vectors (e.g., inner-product, distance) are symmetric, and thus would only be capable of embedding *undirected* graphs. Given a directed graph $G$, one can always first form an undirected graph $G' = (V', E')$ with $2|V|$ nodes, where for each node $i \in V$ we associate two nodes $i_{\text{in}}, i_{\text{out}} \in V'$, and for each edge $(i, j) \in E$ we have an edge $(i_{\text{out}}, j_{\text{in}}) \in E'$. We could embed this graph using symmetric score functions on vectors, for example

$$\mathrm{E}(i_{\text{out}}, j_{\text{in}}) := -\log(\sigma(\phi(i_{\text{out}}) \cdot \phi(j_{\text{in}}))). \tag{1}$$

Of course this can also be retracted to the original directed graph $G$, namely given a parameterization $\phi : V \to \mathbb{R}^{2d}$ we can define an energy function

$$\mathrm{E}_{\text{SIM}}(i,j) := -\log\left(\sigma\left(\phi(i)^T \begin{bmatrix} 0 & I \\ 0 & 0 \end{bmatrix} \phi(j)\right)\right). \tag{2}$$

More generally, for (learned) weights $W \in \mathbb{R}^{2d \times 2d}$ we can define a bilinear score function [37],

$$\mathrm{E}_{\text{BILINEAR}}(i,j) := -\log\left(\sigma\left(\phi(i)^T W \phi(j)\right)\right). \tag{3}$$

## 2.2 Complex Vectors

The models discussed in §2.1 are often considered Euclidean vector embeddings, as they make use of the inner product and linear transformations on Euclidean space. Alternative spaces for representing nodes have been introduced, most notably complex space $\mathbb{C}^d$ [57], where the parametrization can be thought of as $\psi : V \to \mathbb{C}^d$ with energy function

$$\mathrm{E}(i,j) := -\log\left(\sigma\left(\mathrm{Re}\left(\sum_{k=1}^d \psi(i)_k \mathbf{w}_k \overline{\psi(j)_k}\right)\right)\right) \tag{4}$$

where $\mathbf{w} \in \mathbb{C}^d$ and $\mathrm{Re}$ projects to the real part. In order to compare this method to previous models, we can also think of this as parameterized by $\phi : V \to \mathbb{R}^{2d}$, where $\phi(i)_k := \mathrm{Re}\,\psi(i)_k$ and $\phi(i)_{d+k} := \mathrm{Im}\,\psi(j)_k$, in which case the energy function defined as

$$\mathrm{E}_{\text{COMPLEX}}(i,j) := -\log\left(\sigma\left(\phi(i)^T W \phi(j)\right)\right) \quad \text{with} \quad W := \begin{bmatrix} \mathrm{diag}(\mathrm{Re}\,\mathbf{w}) & \mathrm{diag}(\mathrm{Im}\,\mathbf{w}) \\ -\mathrm{diag}(\mathrm{Im}\,\mathbf{w}) & \mathrm{diag}(\mathrm{Re}\,\mathbf{w}) \end{bmatrix} \tag{5}$$

is equivalent to (4).

## 2.3 Hyperbolic Vectors

While the use of bilinear products or complex vectors provide a naturally asymmetric scoring function, one might reasonably ask if there are spaces whose geometry lends itself to representing graphs. It is known, for example, that (undirected) trees can be embedded with arbitrarily low distortion into the Poincaré disk [49], which is not possible in Euclidean space even using an unbounded number of dimensions [31]. These statements are related to *undirected* graphs, and while the in/out parameterization approach taken in the SIM model could be applied here we would lose any geometric benefit, as the undirected graph $G'$ corresponding to a tree is not, itself, a tree. Nickel and Kiela [39] suggest that edge direction can be decided by use of the Euclidean norm, and define a score function for an edge $(u, v)$ as

$$\mathrm{score}(u,v) = -(1 + \alpha(\|u\| - \|v\|))d(u,v), \tag{6}$$

where $d$ is the Poincaré distance and $\alpha$ is a hyperparameter indicating the strength of the penalty for edge directions which violate the order of their norms. An inspection of this score function shows that it is very sensitive to the choice of $\alpha$, which makes training unstable (see Appendix A).

Poincaré distance, in general, suffers from numeric stability issues which have been addressed in more recent work. Nickel and Kiela [40] consider the Lorentz model of hyperbolic space on the

hyperboloid, and Law et al. [28] demonstrates that squared Lorentzian distance on the hyperboloid has similar geometric properties to Poincaré distance, while also providing numerically stable gradients.

Using the notation of Law et al. [28], the hyperboloid is defined as

$$\mathcal{H}^{d,\beta} := \{\mathbf{a} = (a_0, \ldots, a_d) \in \mathbb{R}_+ \times \mathbb{R}^d : \|\mathbf{a}\|_{\mathcal{L}}^2 = -\beta\}, \tag{7}$$

where $\beta > 0$, $\|\mathbf{a}\|_{\mathcal{L}}^2 = \langle \mathbf{a}, \mathbf{a} \rangle_{\mathcal{L}}$ is the *squared Lorentzian norm* of $\mathbf{a}$, and $\langle \mathbf{a}, \mathbf{b} \rangle_{\mathcal{L}}$ is the *Lorentzian inner product*, and the *squared Lorentzian distance* $d_{\mathcal{L}}^2 : \mathcal{H}^{d,\beta} \times \mathcal{H}^{d,\beta} \to \mathbb{R}_+$ is defined as

$$d_{\mathcal{L}}^2(\mathbf{a}, \mathbf{b}) := -2\beta - 2\langle \mathbf{a}, \mathbf{b} \rangle_{\mathcal{L}}, \qquad \langle \mathbf{a}, \mathbf{b} \rangle_{\mathcal{L}} := -a_0 b_0 + \sum_{i=1}^d a_i b_i. \tag{8}$$

Note that $\langle \mathbf{a}, \mathbf{b} \rangle_{\mathcal{L}} \leq -\beta$ for $\mathbf{a}, \mathbf{b} \in \mathcal{H}^{d,\beta}$, with equality if and only if $\mathbf{a} = \mathbf{b}$. We also make use of the bijection $f_\beta : \mathbb{R}^d \to \mathcal{H}^{d,\beta}$, where $f_\beta(\mathbf{x}) := \left( \sqrt{\|\mathbf{x}\|^d + \beta}, x_1, \ldots, x_d \right) \in \mathcal{H}^{d,\beta}$. Given some parameterization $\phi : V \to \mathbb{R}^d$, we define a new hyperbolic energy function:

$$\mathrm{E}_{\text{HYPERBOLIC}}(i, j) := d_{\mathcal{L}}^2(f_\beta \circ \phi(i), f_\beta \circ \phi(j)) + \alpha \log(1 + \exp(\|\phi(i)\| - \|\phi(j)\|)), \tag{9}$$

where the second term serves as a penalty on Euclidean norms to provide a measure of edge direction, i.e., edges should go from nodes with smaller Euclidean norm to larger. (See Appendix Figure 3)

## 2.4 Order Embeddings

An *order embedding* is a mapping between partially ordered sets which preserves the ordering [58]. This relates to our task, as there is a bijection between transitively-closed DAGs and posets. The Order Embedding (OE) model proposed in Vendrov et al. [58] represents elements using vectors in $\phi : V \to \mathbb{R}_+^d$, where a partial order is provided via the reversed product order [3]: $\mathbf{x} \preceq \mathbf{y} \iff \bigwedge_{k=1}^d x_k \geq y_k$. These representations can be thought of as infinite cones $\mathrm{Cone}(\mathbf{x}) := \{\mathbf{y} : \mathbf{x} \preceq \mathbf{y}\}$, in which case $\mathbf{x} \preceq \mathbf{y}$ if and only if $\mathrm{Cone}(\mathbf{x}) \subseteq \mathrm{Cone}(\mathbf{y})$. We consider a model which uses the energy function introduced in Vendrov et al. [58], which provides a smooth penalty for cone containment:

$$\mathrm{E}_{\text{OE}}(i, j) := \|\max(0, \phi(i) - \phi(j))\|_{\ell^2}^2. \tag{10}$$

## 2.5 Hyperbolic Entailment Cones

Ganea et al. [20] extend the ideas of OE to hyperbolic space, modeling nodes as cones in hyperbolic space. This combines the increased capability of hyperbolic space to represent tree-like graphs with the asymmetry and inductive bias of region-based representations. To do so, the parameterization $\phi(i)$ is iterpreted as the apex of a cone in the Poincaré disk, $\mathbb{D}^d$. Ganea et al. [21] define the following functions:

$$\psi(x) = \arcsin\left( K \frac{1 - \|x\|^2}{\|x\|} \right), \quad \Xi(x, y) = \arccos\left( \frac{\langle x, y \rangle (1 + \|x\|^2) - \|x\|^2 (1 + \|y\|^2)}{\|x\| \cdot \|x - y\| \sqrt{1 + \|x\|^2 \|y\|^2 - 2\langle x, y \rangle}} \right)$$

where $\psi(x)$ is the angle of the cone with apex $x$, and $\Xi(x, y)$ is the angle between the half lines $(xy$ and $(0x$. In alignment with our energy interpretation, the energy is given by

$$\mathrm{E}_{\text{HEC}}(i, j) := \max(0, \psi(\phi(i) - \Xi(\phi(i), \phi(j))). \tag{11}$$

## 2.6 Probabilistic Order Embeddings

Lai and Hockenmaier [27] extend OE with a probabilistic interpretation where the volume of the cone under the negative exponential measure represents a marginal probability, $P(i) = \exp\left( -\sum_{k=1}^d \phi(i)_k \right) = \int_{\mathrm{Cone}(\phi(i))} e^{-z} \, dz$. The set of all cones are closed under intersection, i.e. $\mathrm{Cone}(\mathbf{x}) \cap \mathrm{Cone}(\mathbf{y}) = \mathrm{Cone}(\max(\mathbf{x}, \mathbf{y}))$ where $\max$ is the coordinate-wise maximum, thus we can easily calculate the joint probability between elements as $P(i, j) =$

---

[3] As originally defined in Vendrov et al. [58] and Lai and Hockenmaier [27], the vector parameterization for OE and POE was restricted the positive orthant, $\mathbb{R}_+^d$, however in practice one can use $\mathbb{R}^d$, as the energy is a function of the difference between vectors which is preserved under translation.

$\exp\left(-\sum_{k=1}^{d}\max(\phi(i)_k,\phi(j)_k)\right)$. The energy function is $-\log P(i\mid j)$, which simplifies to

$$\mathrm{E}_{\mathrm{POE}}(i,j)\coloneqq\|\max(0,\phi(i)-\phi(j))\|_{\ell^1}. \tag{12}$$

While seemingly a small difference to OE (10), the probabilistic semantics available to POE have ramifications on its ability to model DAGs, as we will discuss in §4.1.

## 2.7 Probabilistic Box Embeddings

Probabilistic box embeddings [60] represent elements with $d$-dimensional hyper-rectangles or *boxes*,

$$\prod_{k=1}^{d}\left[x_k^-,x_k^+\right]=\left[x_1^-,x_1^+\right]\times\cdots\times\left[x_d^-,x_d^+\right]\subseteq\mathbb{R}^d, \tag{13}$$

where $x_k^- < x_k^+$. Let $\mathcal{B}^d$ be the set of all such boxes, along with the empty set. In our notation, these parameters would be provided by $\phi:V\to\mathbb{R}^{2d}$, however for notational convenience we will instead write $\mathrm{Box}:V\to\mathcal{B}^d$. As with cones, boxes are closed under intersection and their volumes can be interpreted as probabilities. Namely, by normalizing the space, we have

$$P(i)=\mathrm{Vol}(\mathrm{Box}(i)),\quad P(i,j)=\mathrm{Vol}(\mathrm{Box}(i)\cap\mathrm{Box}(j)),\quad P(i\mid j)=\tfrac{\mathrm{Vol}(\mathrm{Box}(i)\cap\mathrm{Box}(j))}{\mathrm{Vol}(\mathrm{Box}(j))}.$$

Boxes which are disjoint or entirely contained in one another can present problems for learning algorithms, and thus for training box embeddings we use the approach introduced by Dasgupta et al. [17], wherein latent Gumbel noise is added to the box parameters and the previous volume calculation is approximated using a ratio of (analytically calculable and differentiable) expected box volumes, which results in the following energy function:

$$\mathrm{E}_{\mathrm{Box}}(i,j)\coloneqq\log\mathbb{E}[\mathrm{Vol}(\mathrm{Box}(j))]-\log\mathbb{E}[\mathrm{Vol}(\mathrm{Box}(i)\cap\mathrm{Box}(j))]. \tag{14}$$

# 3 Probabilistic Box Embeddings with Learned Temperatures

In this section, we define our novel box embedding method. As box volume is calculated by taking a product over dimensions, it is sufficient to describe how the calculation is performed in one dimension. Given boxes $[x^-,x^+]$ and $[y^-,y^+]$, note that the volume of intersection is calculated as

$$\mathrm{Vol}([x^-,x^+]\cap[y^-,y^+])=\max(\min(x^+,y^+)-\max(x^-,y^-),0) \tag{15}$$

The difficulty of training an objective involving hard $\min/\max$ functions has been addressed previously [17, 30, 60], but for our purposes we will take a practical approach which is standard for deep learning and replace these hard $\max$ operators with a smooth approximation, namely

$$\mathrm{LogSumExp}(\mathbf{x};t)=t\log(\textstyle\sum_i e^{x_i/t})=\mathrm{LSE}_t(\mathbf{x}), \tag{16}$$

where $t$ is the *temperature* parameter and $\mathrm{LSE}_t(\mathbf{x})\to\max(\mathbf{x})$ as $t\to 0$. Thus we can approximate

$$\mathrm{Vol}([x^-,x^+]\cap[y^-,y^+])\approx\mathrm{LSE}_{t_V}(-\mathrm{LSE}_{t_I}(-x^+,-y^+)-\mathrm{LSE}_{t_I}(x^-,y^-),0), \tag{17}$$

where we call $t_I$ the *intersection temperature* and $t_V$ the *volume temperature*, as this can be thought of as the composition of approximate intersection and volume functions,

$$\mathrm{I}_{t_I}(x^\pm,y^\pm)\coloneqq[\mathrm{LSE}_{t_I}(x^-,y^-),-\mathrm{LSE}_{t_I}(-x^-,-y^-)],\qquad \mathrm{V}_{t_V}(x^\pm)\coloneqq\mathrm{LSE}_{t_V}(x^+-x^-,0). \tag{18}$$

These temperature parameters may be tuned as global hyperparameters, or they may be trained. In our model, we propose to learn a distributed representation for these hyperparameters. Formally, our model parameterizes each node $i$ via $\phi(i)\in\mathbb{R}^{4d}$, which we notionally decompose as

$$\phi(i)=:[\phi(i)^-,\phi(i)^+,t_I(i),t_V(i)], \tag{19}$$

where $\phi(i)^-$ is the min coordinate of a box, $\phi(i)^+$ the max coordinate, $t_I(i)$ the intersection temperature and $t_V(i)$ the volume temperature, each of which are vectors in $\mathbb{R}^d$. We define the energy as

$$\mathrm{E}_{\mathrm{T\text{-}Box}}(i,j)\coloneqq\log\mathrm{V}_\nu(\phi(j)^\pm)-\log\mathrm{V}_\nu(\mathrm{I}_\tau(\phi(i)^\pm,\phi(j)^\pm)) \tag{20}$$

where $\tau=(t_I(i)+t_V(j))/2$ and $\nu=(t_V(i)+t_V(j))/2$. (In higher dimensions, the product over dimensions which results from the volume calculation amounts to summing (20) over dimensions.)

## 3.1 Connection to BOX

We note that the form of this approximation bears a striking similarity to that of the BOX model, which uses the approximation to expected volume derived in [17],

$$\mathbb{E}\left[\max\left(\min(X^+, Y^+) - \max(X^-, Y^-), 0\right)\right]$$
$$\approx \mathrm{LSE}_\beta\left(-\mathrm{LSE}_\beta(\mu_{X^-}, \mu_{Y^-}) - \mathrm{LSE}_\beta(\mu_{X^+}, \mu_{Y^-}) - 2\gamma, 0\right),$$

where the $\mu$ values are the location parameters of the respective Gumbel distributions, the $\beta$ parameters are the variance, and $\gamma$ is the Euler-Mascheroni constant which arises as a consequence of parameterization via the mode rather than the mean. In practice, [17] is trained by tuning separate (global) $\beta$ hyperparameters for intersection and volume, whereas in our model these temperatures are learned, using a per-node, per-dimension distributed representation. This similarity allows for some light comparisons and intuitions to be drawn from the latent random variable perspective provided by the GumbelBox model, for example we can interpret the learned temperatures $t_I(i)$ and $t_V(i)$ as capturing some sense of *uncertainty* related to min and max coordinates for the box representing node $i$. Our model thus represents each node with it's own *position*, *scale*, and *uncertainty*, all of which are learned from data, whereas [17] has a fixed global level of uncertainty. The additional representational capacity of our model matches our intuition that some nodes will be best represented with more or less uncertainty than others, and that uncertainty should start high and generally decrease throughout training. Our intuition is borne out experimentally, where we observe that this allows far more flexibility of the model and, for a given embedding dimension $d$, is always preferable to tuning these temperatures as global hyperparameters (see §6.2).

## 4 Theoretical Analysis

In this section we explore each region-based model's theoretical capability to represent a DAG. We consider the adjacency matrix which results from applying a global threshold $\tau$ to the energy function, i.e. $A_{i,j} = \mathbb{I}[E(i,j) \leq \tau]$. In the following, let $\mathcal{U}$ be the set of all undirected graphs, $\mathcal{G}$ the set of all directed graphs and $\mathcal{D}$ the set of all DAGs. Given a directed graph $G \in \mathcal{G}$, we denote $G_u \in \mathcal{U}$ the undirected graph such that $\{i, j\} \in G_u \Leftrightarrow (i, j) \in G$ or $(j, i) \in G$. In all cases, we will only consider *simple* graphs, that is graphs without multiple edges or self-loops, and remove such edges from the thresholded graph, i.e. set $A_{i,i} = 0$. We define $\mathcal{G}_{\mathrm{M}}(n, \tau) \subseteq \mathcal{G}$ as the set of all graphs with adjacency matrices realizable as the thresholded energy function of model M. That is, $G = (V, E) \in \mathcal{G}_{\mathrm{M}}(n, \tau)$ if and only if there exists some setting of parameters $\phi : V \to \mathbb{R}^n$ such that $E_{\mathrm{M}}(i, j) \leq \tau \iff (i, j) \in E$. We also define $\mathcal{G}_{\mathrm{M}}(n) := \bigcup_{\tau \in \mathbb{R}_+} \mathcal{G}_{\mathrm{M}}(n, \tau)$ and $\mathcal{G}_{\mathrm{M}} := \bigcup_{n \in \mathbb{N}} \mathcal{G}_{\mathrm{M}}(n)$.

### 4.1 Asymmetrization

An initial observation is that $\mathcal{G}_{\mathrm{M}}$ is not necessarily a subset of $\mathcal{D}$. For example, it may be the case for some parameterization that $E_{\mathrm{M}}(i, j) \leq \tau$ and $E_{\mathrm{M}}(j, i) \leq \tau$. We can address this issue with one additional constraint: let $\mathcal{G}_{\mathrm{M}}^{\mathrm{ASYM}} \subseteq \mathcal{G}$ be the set of graphs $G$ where edge $(i, j)$ exists if and only if

$$E_{\mathrm{M}}(i, j) \leq \tau \text{ and } E_{\mathrm{M}}(i, j) < E_{\mathrm{M}}(j, i). \tag{21}$$

This approach was adopted in Vilnis et al. [60], where it was proved that $\mathcal{G}_{\mathrm{M}}^{\mathrm{ASYM}} \subseteq \mathcal{D}$ for any model which uses conditional probabilities for the energy function.

**Proposition 1** (Vilnis et al. [60]). *Suppose $E_{\mathrm{M}}(i, j) = -\log P(X_i \mid X_j)$ for some probability distribution $P$ over binary random variables $\{X_i\}_{i \in V}$, then $\mathcal{G}_{\mathrm{M}}^{\mathrm{ASYM}} \subseteq \mathcal{D}$.*

For completeness, we include a version of this proof in Appendix B. The following is an immediate consequence.

**Corollary 1.** *The asymmetrization of Probabilistic Order Embeddings, which use energy function $E_{\mathrm{POE}}(i, j) = \|\max(0, \phi(i) - \phi(j))\|_{\ell^1}$, will always yield a DAG, that is $\mathcal{G}_{\mathrm{POE}}^{\mathrm{ASYM}} \subseteq \mathcal{D}$.*

A direct proof of this fact is also provided in Appendix B. An interesting observation is that this same condition is not sufficient for all methods, in particular it does not apply to OE, despite the similarity in energy function:

**Proposition 2.** *The asymmetrized adjacency matrix for order embeddings, with energy* $\mathrm{E_{OE}}(i, j) = \|\max(0, \phi(i) - \phi(j))\|_{\ell^2}^2$, *may not yield an acyclic graph, i.e.* $\mathcal{G}_{\mathrm{OE}}^{\mathrm{ASYM}} \not\subseteq \mathcal{D}$.

The proof can be found in Appendix B.

## 4.2 Representational Capacity

The statements in the previous section discuss when it is possible to, for any parameters in a given model, extract a DAG by thresholding the energy function and asymmetrizing. The arguably more interesting question, however, is whether a given model can represent *all* DAGs, i.e. do we have $\mathcal{D} \subseteq \mathcal{G}_{\mathrm{M}}$? Bhattacharjee and Dasgupta [4] refer to the SIM model as a *similarity embedding*, and prove that it can represent any graph, and by extension so can BILINEAR. Here we provide a proof that box embeddings can represent any DAG, i.e. $\mathcal{D} \subseteq \mathcal{G}_{\mathrm{Box}}$.

The concept of *boxicity* of an undirected graph $G = (V, E) \in \mathcal{U}$ was introduced by Roberts [45] as the minimum $d$ for which there exists a parameterization $\mathrm{Box} : V \to \mathcal{B}^d$ such that $\{i, j\} \in U$ if and only if $\mathrm{Box}(i) \cap \mathrm{Box}(j) \neq \emptyset$. Roberts [45] also proves that the boxicity of any graph is at most $\lfloor |V|/2 \rfloor$. We leverage this result to prove the following.

**Theorem 2.** *Given any* $G = (V, E) \in \mathcal{D}$, *there exists a parameterization* $\mathrm{Box} : V \to \mathcal{B}^d$ *and threshold* $\tau$ *such that* $\mathrm{E_{T\text{-}BOX}}(i, j) < \tau$ *if and only if* $(i, j) \in E$.

*Proof Sketch* We first obtain a boxicity representation for the undirected graph $G_u = (V_u, E_u)$, for which $i, j \in E_u$ if $(i, j) \in E$ or $(j, i) \in E$. In this representation, $Box(i) \cap Box(j) \neq \emptyset$ implies that either $(i, j) \in E$ or $(j, i) \in E$, and thus we simply must adjust the marginal volumes to represent the correct direction without disrupting the existing overlaps. We can by extending into one more dimension, and assigning heights of the boxes in this new dimension in accordance with a topological sort. For full details, see the proof in Appendix C.

As our proof is constructive, existing boxicity algorithms can be extended to represent any directed graph in this manner, which also serves to provide dimensionality bounds. For example, Chandran et al. [14] provide a randomized algorithm to generate a boxicity representation of an undirected graph $U = (V, E)$ in $\mathcal{O}((\Delta + 2) \log |V|)$ dimensions where $\Delta$ is the maximum degree of $U$. We include a version of this randomized algorithm in Appendix E, where slight modifications were made to guarantee a positive box volume and avoid boxes touching each other but not overlapping.

These theoretical statements apply in the limit where the box temperatures $\to 0$, however the fact that the temperatures are trainable in T-BOX makes this setting feasible, and further is frequently observed empirically during training.

## 4.3 Precision

It is well known that representation results may trade-off dimensions for an increase in precision. For example, Kang and Müller [24] show that embedding an undirected graph using vectors in minimal dimension could require $\mathcal{O}(e^{|V|})$ bits. For this reason, we provide an upper bound on the required bits for the constructive proof.

**Proposition 3.** *Given* $G \in \mathcal{D}$, *there exists a box embedding of* $G$ *in dimension* $\mathcal{O}((\Delta + 2) \log |V|)$ *requiring* $\mathcal{O}(D(\Delta + 2) \log^2 |V|)$ *bits per box, where* $D \leq |V|$ *is the depth of* $G$, *and* $\Delta$ *the maximum degree.*

*Proof Sketch.* We consider each dimension of the boxes in the construction for theorem 2. For all but the last dimension, we can use integer coordinates and preserve box overlaps. This is the boxicity representation, and thus requires $d = \mathcal{O}((\Delta + 2) \log |V|)$ dimensions, thus the total precision required for these dimensions is $\mathcal{O}((\Delta + 2) \log^2 |V|)$. For the last dimension, we can choose a threshold $\tau = 2^{-\lceil d \log(2|V|) \rceil}$. By construction, $h(j) = \tau^k$ for some $k = \{0, \ldots, D\}$, thus we conclude that this dimension requires $\mathcal{O}(Dd \log |V|)$ bits. See Appendix F for a complete proof.

## 5 Objectives for Learning

We fit geometric embeddings by optimizing a binary cross entropy objective, interpreting the energy functions described in §2 and §3 as negative log probability of edge existence, and compare the F1

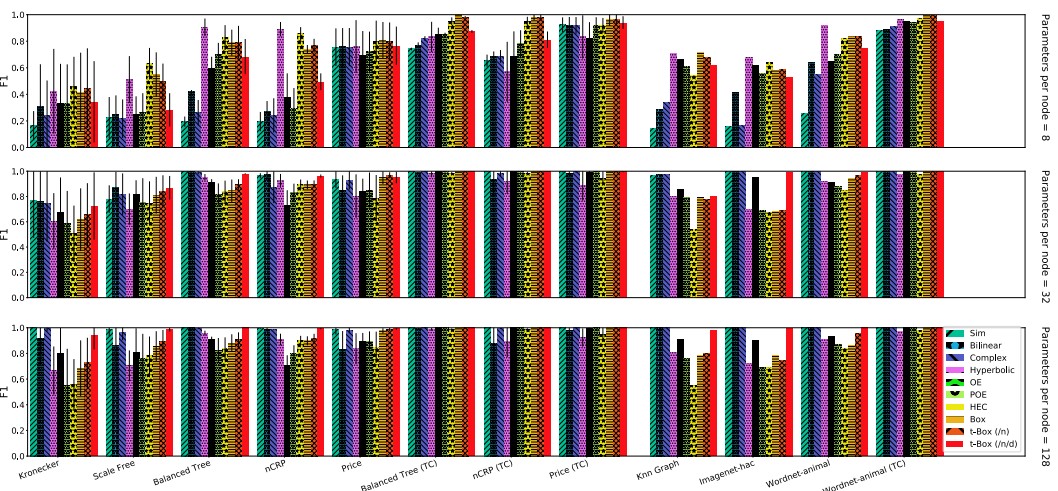

Figure 1: **Representational Capacity** We plot the F1 score for each embedding method using 8, 32, and 128 parameters per node. TC indicates that the transitive closure is applied to the graph. /n and /d indicate box models have per node / per dimension learned temperatures.

score using an optimal threshold. We sample minibatches of positive edges; for each positive edge we sample 128 (true) negative examples uniformly at random, and use a self-adversarial negative weight, as described in [54]. For $E_{OE}$ and $E_{HEC}$, to remain consistent with the originals, we use the loss function as described in Vendrov et al. [58]. Additional details are provided in Appendix D.

# 6 Experiments

In this section, we empirically analyze the following questions:

- How well can each of the embedding models, fit with gradient descent, represent a given graph?
- How does the model's embedding dimensionality or number of parameters impact performance?
- What characteristic properties of graphs are indicative of performance of the embedding models?

To test hypotheses regarding representational capacity and inductive biases, we evaluate on a wide variety of both synthetic and real-world graphs. To avoid optimization difficulties, all embedding models are extensively tuned using Bayesian hyperparameter optimization.

## 6.1 Experimental Setup

**Graphs - Synthetic.** We select five generative models of graphs to build synthetic datasets, aiming for diversity in the kinds of graphs used for evaluation. Each generative model has hyperparameters that control the particular structure of the generated graph. These include: **Balanced Trees** (branching factor $b$); **the Nested Chinese Restaurant Process**, [6] (normalized "new table" probability $\alpha$), which is a non-parameteric tree structure with rich-get-richer properties; **Price's model** [42] (num. connections for new node, $m$, probability of vertex receiving an edge, $c$), which simulates growth of citation networks with preferential attachment and is the directed version of the Barabási–Albert model [1]; **Scale-Free Networks** [7] ($\alpha, \beta, \gamma, \delta_{in}$) exhibit a power-law degree distribution. We consider a directed scale-free model for which both the in- and out-degree distributions follow power-law distributions; **Kronecker graphs** [29] (initial seed matrix $\{a, b; c, d\}$ and the Kronecker power $k$) are a generative model for graphs which provably exhibit many of the properties of real-world graphs. Please refer to Appendix G for more details.

**Graphs - Real World.** We also select the following real world graph datasets: **WordNet (Animals)** [34]. In our experiments, we took the subtree of "animal" which contains 4,017 nodes and 4,051 direct edges as well as the transitive closure (29,795 edges). **Hierarchical Clustering**. We run agglomerative clustering on the InceptionV3 [56] features from $2^{13}$ ImageNet images [47]. Using the same features we also build and use a **K-Nearest Neighbor Graph** (using K=5).

**Evaluation Metric for Representational Capacity.** Representational capacity is a measure of reconstruction performance, *not generalization*. Given a graph $G^\star$, each model is fit on $G^\star$ and predicts a graph $\hat{G}$. We report the F1 measure between the edges of $G^\star$ and $\hat{G}$ to measure performance.

**Hyperparameter Optimization.** To remove hyperparameters as confounding factors to the model's performance (and inferred representational capacity), we perform extensive Bayesian hyperparameter optimization using W&B [5]. For each of the generative models, we consider a variety of settings for generating samples (see Appendix, Table 1). For each setting $\theta \in \Theta$, we sample 10 graphs $G_\theta$, and then run a dedicated hyperparameter search for each embedding method to maximize the expected performance over sampled graphs. Then for each $G_\theta$, we choose the best F1 achieved in the hyperparameter search and we report the average performance over all 10 sampled graphs from a type of generative model. In total we have 58 hyperparameter settings over these five types of graphs, which leads to 508 graph samples. For each real world graph, we also perform a hyperparameter search. In both cases we calculate F1 over the full adjacency matrix.

**Model details.** We empirically compare each of the geometric embeddings and energy functions described in Section 2: : $E_{\text{SIM}}$, $E_{\text{BILINEAR}}$, $E_{\text{COMPLEX}}$, $E_{\text{HYPERBOLIC}}$, $E_{\text{OE}}$, $E_{\text{POE}}$, $E_{\text{BOX}}$, and the proposed learned-temperature box model $E_{\text{T-BOX}}$. All models are tuned on learning rate, batch size, and weight of negative loss. We tune the margin $\gamma$ for OE, $\alpha$ and $\beta$ parameters in (9) for HYPERBOLIC, intersection and volume temperature for BOX, and the initialization of these temperatures for T-BOX. We check the training loss ten times per epoch, and apply early-stopping with a patience of just over 2 epochs (21 loss observations).

## 6.2 Results & Analysis

Figure 1 reports the mean (and for synthetic graphs, standard deviation) of F1 performance for each model on synthetic and real world graphs, with 8, 32, 128 parameters per node (*not* dimension).

**Observation: Geometric embeddings excel over vectors in low dimensions.** In the lowest dimensions, the BOX model achieves the highest average F1 on the datasets except the two tree-based datasets (nCRP and balanced trees without transitive closure). Hyperbolic is top performing on the tree-based datasets, consistent with previous work. Vector-based approaches perform significantly worse in lower dimensions. This is consistent with the intuition that vectors have flexible representation capacity but lack the inductive biases for better performance in lower dimensions. Further, the hyperbolic methods tend to struggle to on transitively closed graphs.

**Observation: T-BOX out-performs other geometric models in higher dimensions** The flexibility of the vector-based approaches (i.e. SIM, COMPLEX, BILINEAR) can more easily fit graphs with more parameters/higher dimensions. However, previous geometric embedding models do not observe as rapid a performance increase as we increase the dimension. In comparison, our proposed T-BOX can significantly improve the performance on larger dimensions and leads to best overall performance in both high and low dimension space.

**Observation: T-BOX consistently improves BOX for fixed dimension.** While T-BOX(/N/D) generally outperforms BOX, we observe that when there are 8 parameters per node T-BOX(/N/D) achieves lower F1 scores than BOX. With 8 parameters per node, the BOX has 4 dimensional boxes and the T-BOX(/N/D) has 2 dimensional boxes. The reduction to two dimensional boxes restricts the capacity. We subsequently analyze the performance of the models as a function of dimension (rather than number of parameters). In Figure 2, we verify that learning the temperature improves expressiveness by comparing BOX, T-BOX, T-BOX(/D), T-BOX(/N), T-BOX(/N/D) for various dimensionalities. Results indicate that a T-BOX(/N/D) always out-performs original BOX.

**Observation: Relationship between Graph Characteristics & Model Performance.** To get better insight about why certain graphs are easier/harder for methods to embed we calculate graph statistics, including sparsity; clustering coefficient; reciprocity (i.e. proportion of reverse edges); flow hierarchy (fraction of edges not in cycles). We are interested in understanding which graph properties are representable by which models. In Appendix I, we show parallel coordinate plots among different models and calculate Spearman correlation $\rho$ between relative F1 and graph characteristics. These reveal that: box embeddings tend to prefer graphs with lower degree, lower clustering coefficient, higher flow hierarchy and lower reciprocity, indicating these geometric embeddings can better model directed graphs with less cycles and lower density.

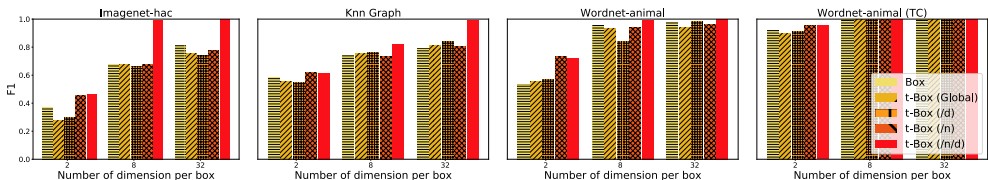

Figure 2: **Constant temperature vs Learning the temperature** In this plot we compare performances between original Gumbel box and variations of T-Box under a same dimension.

# 7 Related Work

Our work intersects several related areas of machine learning including: distributed representations, probabilistic models of graphs, representations of graphs and knowledge-graph link prediction.

**Geometric Embeddings**. Using geometrically shaped embedded representations instead of simple vector-based models have considered Gaussian distributions [59], cones in both Euclidean [58] and hyperbolic space [20], and boxes [53]. Region-based embeddings whose representation can be assigned a volume under some finite measure can also be normalized to represent a probability distribution, as has been done with cones [27] and boxes [60]. Recent work also seeks to improve the learning capability of geometric embeddings. In the hyperbolic setting, new parameterizations which allow for more efficient Riemannian optimization [40] or alternative distance functions [28] have been proposed. For box embeddings, improvements to training have been introduced by smoothing the objective via convolution [30] or by using a latent noise model with a closed-form solution [17]. In this work, to fairly compare all methods, we take these recent improvements into account.

**Representing Graphs**. Embedded representations of graphs has also been widely studied [8, 11, 15, 23, 28, 32, 40, 38, 41, 57, 61, 63]. Embedding tree structures in hyperbolic space both algorithmically [48, 49] as well as multi-dimensional scaling [48]. Hyperbolic space has also been used in representing tree structured latent variables [33, 51] and for hierarchical clustering [12, 35]. Most similar to our work, Weber [61] analyzes the representational capacity of different geometries (Euclidean, hyperbolic, spherical) on undirected graphs, however this is concerned with the *distortion* of a graph, i.e., attempting to preserve distances between nodes, whereas our work investigates the ability of a graph to perform edge classification.

**Knowledge Graph Embeddings**. The representation of directed graphs and prediction of missing edges has been widely studied in the setting of completing missing information in knowledge graphs [8, 13, 22, 26, 36, 44, 52, 54, 64]. Many novel representations have been created for this purpose, such as complex [57] or quaternion [65] vector representations. The task setting differs from our work in various ways - the graph can be viewed as an multi-graph with labeled edges, and the objective is to recover missing edges - however one of the main goals of our work is to provide insight into the inductive biases present in the geometric embeddings which can be used for such a task.

**Probabilistic Models**. Probabilistic models has been studied for both graph generation and link prediction. Probabilistic generative models of graphs are widely studied to generate undirected graphs such as Erdos-Rényi [18], Barabási-Albert [3] and directed graphs such as Price's Model [42], Kronecker [29] along with others [6, 9, 10, 16]. There are also Bayesian non-parametric models for link prediction in directed and relational graphs [19, 25, 46].

# 8 Conclusion

In this paper, we examine the representational capacity and inductive biases of learned representations of graphs. We theoretically and empirically analyze geometric embedding methods for modeling directed graphs. We prove theoretical statements about order embeddings, probabilistic order embeddings, and box embeddings. We propose a new box embedding model that learns temperatures which out-performs other geometric representations in low dimension and shows competitive performance with vector-based models in high dimension. We perform an empirical comparison of seven methods on both synthetic datasets and real world datasets. Our work describes meaningful theoretical foundations and empirical analysis of geometric representations of directed graphs.

**Acknowledgements**

This work supported in part by the Center for Data Science and in part the Center for Intelligent Information Retrieval, and in part by the National Science Foundation under Grants No. 1763618, University of Southern California subcontract no. 123875727 under Office of Naval Research prime contract no. N660011924032, and in part by IBM Research AI through AI Horizons Network, and in part by the Chan Zuckerberg Initiative under the project "Scientific Knowledge Base Construction". The U.S. Government is authorized to reproduce and distribute reprints for Governmental purposes notwithstanding any copyright notation thereon. Some of the work reported here was performed using high performance computing equipment obtained under a grant from the Collaborative R&D Fund managed by the Massachusetts Technology Collaborative. Any opinions, findings and conclusions or recommendations expressed in this material are those of the authors and do not necessarily reflect those of the sponsor.

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
