# A Hyperbolic Model

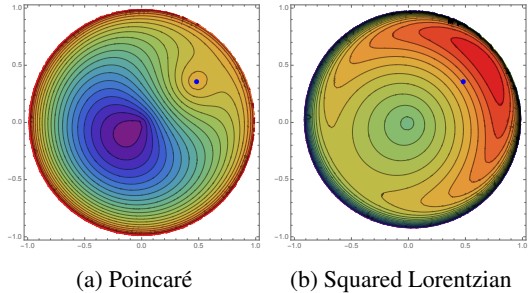

(a) Poincaré      (b) Squared Lorentzian

Figure 3: **Hyperbolic Embedding Score Functions** Countour plots depict the score for edge existence, where red indicates high probability of an edge from the blue point. As opposed to our proposed model, the multiplicative score for Poincaré results in values on the opposite side of the space scoring highly as an edge.

# B Asymmetrization Proofs

**Proposition 1** (Vilnis et al. [60]). *Suppose* $E_{\mathrm{M}}(i,j) = -\log P(X_i \mid X_j)$ *for some probability distribution* $P$ *over binary random variables* $\{X_i\}_{i \in V}$, *then* $\mathcal{G}_{\mathrm{M}}^{\mathrm{ASYM}} \subseteq \mathcal{D}$.

*Proof.* Note that the asymmetrization constraint $E_{\mathrm{M}}(i,j) < E_{\mathrm{M}}(j,i)$ is equivalent, in this case, to $P(X_i) > P(X_j)$. Without loss of generality, assume the variables are ordered so that $P(X_i) \geq P(X_j)$ if $i < j$. The resulting adjacency matrix $A$ formed by thresholding according to (21) will therefore be upper-triangular, and by removing the diagonal it will be nilpotent. As the entries of $A^k$ represent $k$-hop neighbors, the fact that $A^k = 0$ for some $k$ implies the graph is acyclic. $\square$

**Corollary 3.** *The asymmetrization of Probabilistic Order Embeddings, which use energy function* $E_{\mathrm{POE}}(i,j) = \|\max(0, \phi(i) - \phi(j))\|_{\ell^1}$, *will always yield a DAG, that is* $\mathcal{G}_{\mathrm{POE}}^{\mathrm{ASYM}} \subseteq \mathcal{D}$.

*Proof.* While this can be viewed as a consequence of 1 We have

$$E_{\mathrm{POE}}(i,j) + \sum_k \phi(j)_k = \sum_k \max(0, \phi(i) - \phi(j)) + \sum_k \phi(j)_k \qquad (22)$$

$$= \sum_k \max(\phi(i), \phi(j)) \qquad (23)$$

$$= E_{\mathrm{POE}}(j,i) + \sum_k \phi(i)_k. \qquad (24)$$

Thus,

$$E_{\mathrm{POE}}(i,j) - E_{\mathrm{POE}}(j,i) = \sum_k \phi(i)_k - \sum_k \phi(j)_k, \qquad (25)$$

which implies

$$E_{\mathrm{POE}}(i,j) > E_{\mathrm{POE}}(j,i) \iff \sum_k \phi(i)_k > \sum_k \phi(j)_k. \qquad (26)$$

Reordering the nodes according to $\sum_k \phi(i)_k$, we see that the asymmetrized adjacency matrix will have nonzero entries only above the diagonal, which is nilpotent, and thus the graph has no cycles. $\square$

**Proposition 2.** *The asymmetrized adjacency matrix for order embeddings, with energy* $E_{\mathrm{OE}}(i,j) = \|\max(0, \phi(i) - \phi(j))\|_{\ell^2}^2$, *may not yield an acyclic graph, i.e.* $\mathcal{G}_{\mathrm{OE}}^{\mathrm{ASYM}} \not\subseteq \mathcal{D}$.

*Proof.* Consider the following set of 3 3-dimensional order embeddings:

$$u_1 = (0.5, 0.6, 0.3)$$
$$u_2 = (0.8, 0.0, 0.7)$$
$$u_3 = (0.2, 0.3, 0.9).$$

This yields the following matrix of pairwise energies,

$$A = \begin{pmatrix} 0 & 0.36 & 0.18 \\ 0.25 & 0 & 0.36 \\ 0.36 & 0.13 & 0 \end{pmatrix}.$$

If we set $C$ equal to the asymmetrized version of $A$, we have

$$C = \begin{pmatrix} 0 & 1 & 0 \\ 0 & 0 & 1 \\ 1 & 0 & 0 \end{pmatrix}.$$

Since this matrix is not nilpotent, the graph has a cycle. $\square$

## C   Proof: Box Can Represent Any DAGs

**Theorem 2.** *Given any $G = (V, E) \in \mathcal{D}$, there exists a parameterization* $\mathrm{Box} : V \to \mathcal{B}^d$ *and threshold $\tau$ such that* $\mathrm{E}_{\text{T-Box}}(i, j) < \tau$ *if and only if* $(i, j) \in E$.

*Proof.* Let $G = (V, E) \in \mathcal{D}$. Without loss of generality, assume the nodes are labeled according to a topological sort (linear extension), such that if $(i, j) \in E$ then $i < j$.

By the result of Roberts [45], there exists a $d$ dimensional box embedding of $G_u$. Scaling the space if necessary, let $P_{\leq d}$ be the probability distribution given by the boxes in these $d$ dimensions. Note that $P_{\leq d}(i, j) > 0$ if and only if $\{i, j\} \in E_u$. Define the threshold

$$\tau := \min_{(i,j) \in E} P_{\leq d}(i \mid j)/2 > 0. \tag{27}$$

Extend the boxes into dimension $d + 1$, so that the projection of $\mathrm{Box}(i)$ in this new dimension is $[0, h(i)]$ where $h : V \to \mathbb{R}_+$ is a strictly positive function to be defined. We use $P$ to denote the probability distribution implied by the box embeddings in the full $d + 1$ dimensions, and note:

i. $P(i) = h(i)P_{\leq d}(i)$

ii. $P(i, j) = \min(h(i), h(j))P_{\leq d}(i, j)$

iii. $P(i, j) > 0$ if and only if $(i, j) \in E$ or $(j, i) \in E$

If node $i$ has no in-edge, then $h(i)$ can be any number at all, say $h(i) = 1$. Assign the remaining values as follows: Let $j$ be the smallest number for which $h(j)$ is not yet defined, and let $S = \{i : (i, j) \in D\}$. Since the nodes are indexed according to topological sort, for all $s \in S$ we have $s < j$ and thus $h(s)$ has been defined.

Let $h(j) := \min_{s \in S} h(s)\tau$, then for any $s \in S$ we have

$$P(s|j) = P_{\leq d}(s|j)\frac{\min(h(s), h(j))}{h(j)} = P_{\leq d}(s|j)\frac{h(j)}{h(j)} > \tau$$

and

$$P(j|s) = P_{\leq d}(j|s)\frac{\min(h(s), h(j))}{h(s)} \leq P_{\leq d}(j|s)\frac{h(s)}{h(s)}\tau \leq \tau.$$

Therefore, the graph generated from this box embedding will have edge $(s, j)$ and will not have $(j, s)$ for all $s \in S$, which completes the proof. $\square$

---

**Algorithm 1:** Randomized Algorithm to Generate Boxicity Representation for Undirected Graphs

---

**Def** $\mathcal{M}$ (*U* $\in \mathcal{U}$, *permutation* $\pi$):

   | **for** $X \in U$ **do**

   |   | Assign interval $I_X = [\pi(X_N), \pi(X) + 0.5]$ where $\pi(X_N) = min_{w \in N[X]}\pi(w)$, and

   |   | $N[X]$ is a set of closed neighborhood of $X$

   | **end**

   | Generate a interval graph $U'$ where $(X_i, X_j) \in U' \Leftrightarrow |I_{X_i} \cap I_{X_j}| > 0$

   | **return** $U', I$

**Def** RAND ($U = (V, E) \in \mathcal{U}$):

   | Generates a permutation $\pi$ of $\{1, 2, 3, ..., |V|\}$ uniformly at random

   | $U', I := \mathcal{M}(U, \pi)$

   | **return** $U', I$

**Def** main ($U = (V, E) \in \mathcal{U}$):

   | $d := \lceil (\Delta + 2) \ln |V| \rceil$

   | **repeat**

   |   | **for** $i \in \{1, 2, ..., \}$ **do**

   |   |   | $U'_i, I_i := $ RAND$(U)$

   |   | **end**

   | **until** $U'_1 \cap U'_2 \cap ... \cap U'_d = U$

   | **return** $I_1, I_2, ..., I_d$

---

# D  Objectives for Learning Geometric Representations

We apply binary cross entropy over energies as our loss function (except $\mathrm{E_{OE}}$):

$$L = (1 - w) \, \mathrm{E}(i, j) - w \sum_{(i', j') \in N(i, j)} w(i', j') \log(1 - e^{- \mathrm{E}(i', j')}) \tag{28}$$

where $w$ is the weight of negative loss, $w(i', j')$ is a normalized weight for each negative sample and $N(i, j)$ is the set of negative samples for each positive edge $(i, j)$ within one batch. In practice, we found that the performance can be improved significantly by up sampling more challenging negatives. We convert energy to a logit and then follow self-adversarial negative sampling strategy from Sun et al. [54]:

$$\mathrm{logit}(i', j') = - \mathrm{E}(i', j') - \log(1 - e^{- \mathrm{E}(i', j')}) \tag{29}$$

and

$$w(i', j') = \frac{e^{\mathrm{logit}(i', j')}}{\sum_{(i', j') \in N(i, j)} e^{\mathrm{logit}(i', j')}} \tag{30}$$

We apply Bayesian optimization for hyperparameter search. Table 2 shows the range of hyperparameters we search.

# E  Boxicity Representation for Undirected Graphs

See Alg.(1).

# F  Precision Bound

**Proposition 3.** *Given $G \in \mathcal{D}$, there exists a box embedding of $G$ in dimension $\mathcal{O}((\Delta + 2) \log |V|)$ requiring $\mathcal{O}(D(\Delta + 2) \log^2 |V|)$ bits per box, where $D \leq |V|$ is the depth of $G$, and $\Delta$ the maximum degree.*

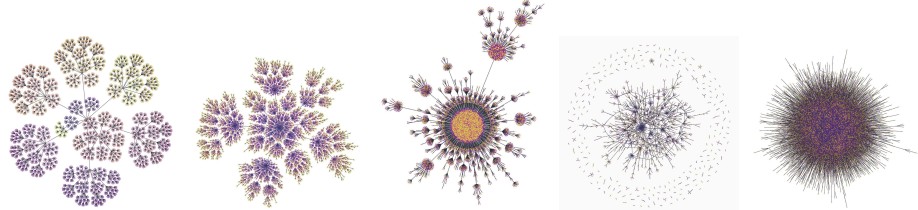

Figure 4: **Examples of each of the families of generative models.** *(left to right)*: Balanced tree, Nested Chinese Restaurant Process, Price's, Kronecker, Scale-Free network

*Proof.* We refine the constructive proof in Theorem 2, and denote box $i$ as

$$\text{Box}(i) = \prod_{k=1}^{d+1} [x_{i,k}^-, x_{i,k}^+] = [x_{i,1}^-, x_{i,1}^+] \times \cdot \times [x_{i,d+1}^-, x_{i,d+1}^+].$$

In the first step, boxes were constructed via boxicity such that

$$P_{\leq d}(i, j) > 0 \iff \{i, j\} \in E_u. \tag{31}$$

Note that Chandran et al. [14] provides a dimensionality bound of $d = \mathcal{O}((\Delta + 2) \ln |V|)$, where $\Delta$ is the maximum degree of $G_u$. The intersection pattern in dimension $k$ is preserved as long as the order of the $2|V|$ numbers in $S_k := \cup_i \{x_{i,k}^-, x_{i,k}^+\}$ is maintained. Thus, for each dimension $k \in \{0, \ldots, d\}$, we can replace the parameters in $S_k$ by their rank and preserve property (31). Each box therefore requires at most $\log(2|V|)$ bits in the first $d = \mathcal{O}((\Delta + 2) \log |V|)$ dimensions.

To handle DAGs, as mentioned in the proof for Theorem 2, we need one more dimension to create marginal probabilities with the same order as the topological sort. Note that, after the above ranking adjustment, the coordinates in the first $d$ dimensions are integers, and so we have

$$\min_{(i,j)\in E} P_{\leq d}(i, j) \geq 1. \tag{32}$$

Furthermore, the largest a single box could be in any given dimension is $2|V|$, and so

$$\min_{(i,j)\in E} P_{\leq d}(i \mid j) \geq (2|V|)^{-d}. \tag{33}$$

Therefore we can choose a threshold

$$\tau = 2^{-\lceil d \log(2|V|) \rceil - 1} < P_{\leq d}(i \mid j). \tag{34}$$

By construction, $h(j) = \tau^k$ for some $k = \{0, \ldots, D\}$, where $D$ is the max depth of the graph, thus we conclude that this dimension requires $\mathcal{O}(Dd \log |V|)$ bits, which completes the proof. $\qquad \square$

# G   Generative Models of Graphs

We would like to consider a wide variety of models of graphs, which have a diverse set of characteristics. By considering a wide variety of generative models for graphs, we would hope to understand where and when each of the proposed approaches is effective.

We consider the following models of graphs:

**Balanced Trees** have a fixed branching factor $b$ at each node. To build a tree with a given number of nodes, we allow the last level structure to be incomplete (i.e., missing nodes) if the number of nodes is not a power of the branching factor. We also convert to the undirected tree to out-tree by letting edges point away from the root.

**The Nested Chinese Restaurant Process** (nCRP) [6] defines a distribution over infinitely wide/deep trees. The nCRP can be described as a recursively applied Chinese Restaurant Process (CRP) at each node in the tree structure. Each descendant leaf node of a particular node $n$ is assigned to one of the $n$'s children, say $c$, with probability proportional to the number of descendants of $c$ and starts a new child of $n$ with some probability (given by hyperparameter $\alpha$). Note that unlike balanced trees, this

| Generative Model | Range of $\Theta$ |
| --- | --- |
| Balanced Tree | $b \in \{2, 3, 5, 10\}$ 
 TC $= \{true, false\}$ |
| nCRP | $\alpha \in \{10, 100, 500\}$, 
 TC $= \{True, False\}$ |
| Price's network | $c \in \{0.1, 0.01\}$, $m \in \{1, 5, 10\}$, 
 TC $= \{True, False\}$ |
| Scale Free Network | $\alpha = \{0.1, 0.3\}$, $\delta_{in} = \{0.0, 1.0\}$, 
 $\delta_{out} = \{0.0, 1.0\}$, $\gamma = \{0.4, 0.6\}$ |
| Kronecker Graph | $a = \{0.8, 1.0\}$, $b = \{0.4, 0.6\}$, 
 $c = \{0.3, 0.5\}$, $d = \{0.1, 0.3\}$ |

Table 1: **Hyperparameter Settings for Synthetic Graph Generation.** "TC" means including full transitive closures as ground truth. In our experiments, the number of nodes is $2^{13}$ for all generative models.

| Hyper-Parameter | Range |
| --- | --- |
| Learning rate | [1e-4, 1.0] |
| batch size | $\{2^8, 2^9, 2^{10}, 2^{11}\}$ |
| $w$ | [0.0, 1.0] |
| $\gamma$ | [0.0, 10.0] |
| $\alpha$ | [0.0, 10.0] |
| $\beta$ | [0.0, 10.0] |
| $t_{intersection}$ | [1e-4, 0.5] |
| $t_{volume}$ | [0.1, 10.0] |

Table 2: **Hyperparameter Range for Bayesian Optimization.**

tree structure will exhibit a much more uneven distribution of the nodes in terms of depth. There are certain subtrees that are very 'heavy', containing many nodes and other subtrees that are quite shallow and only contain a handful of nodes.

**Price's model** [42] simulates the growth of citation networks with preferential attachment, where the probability of an existing paper to get new citations is proportional to the number of existing citations the paper already has. Its undirected version is sometimes also known as the Barabási–Albert model [1]. In this model, each new node will point to $m$ existing vertices, and $c$ is a constant factor added to the probability of a vertex receiving an edge.

**Scale-Free Networks** exhibit a power-law degree distribution. We consider a directed scale-free model for which both the in- and out-degree distributions follow power-law distributions [7]. The generative model has five hyperparameters, $\alpha, \beta, \gamma, \delta_{in}, \delta_{out}$. To generate a graph edges are sampled according to: (1) with probability $\alpha$, create a node $n$, and create one edge from node $n$ and an existing node $m$ (sampled among existing nodes with probability proportional to $\deg_{in}e(m) + \delta_{in}$); (2) with probability $\beta$, create an edge from one existing node $n$ to $m$ (sampled among existing nodes with probability proportional to $\deg_{out}(n) + \delta_{out}$ and $\deg_{in}(m) + \delta_{in}$ respectively); (3) with probability $\gamma$, create a new node $n$ and an edge from an existing node $m$ to $n$ (with $m$ sampled with probability proportional to $\deg_{out}(m) + \delta_{out}$).

**Kronecker graphs** [29] describes a generative model for graphs which provably exhibit many of the properties of real-world graphs (including heavy tailed degree distributions, small diameters, etc). The model generates self-similar graphs recursively based on Kronecker product of given seed matrix. It requires a initial seed matrix $\{a, b; c, d\}$ and the Kronecker power $k$ which determines the number of times Kronecker product will be applied. We use $k = 13$.

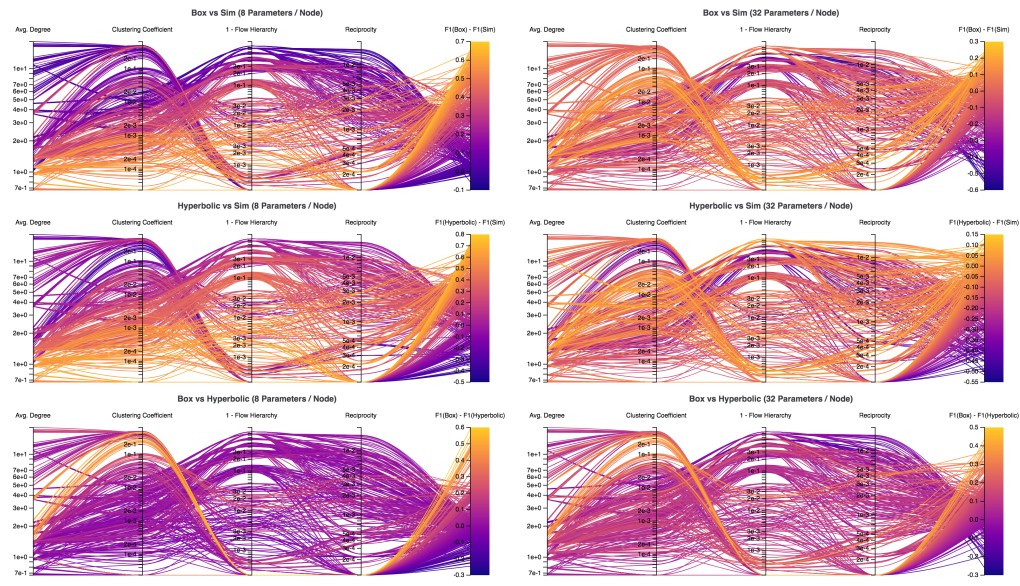

Figure 5: **Parallel Coordinate Plot over Graph Characteristics & Relative F1**. Plots on the left hand side are using 8 parameters per node, and right hand side are using 32 parameters per node.

# H  Real World Graphs

**WordNet** [34] is a database that contains semantic relationships between English nouns, adjectives, verbs and adverbs. Following previous work [39, 60], we consider a graph in which each node is a noun and edges exist from node $i$ to $j$ if $i$ is a hypernym (i.e., "is-a") $j$. In our experiments, we took the subtree of "animal" which contains 4,017 nodes and 4,051 direct edges. And we have 29,795 edges after applying full transitive closure (following the semantics of the hypernym relation).

**Hierarchical Clustering**. We run hierarchical agglomerative clustering on the InceptionV3 [56, 47] features from $2^{13}$ ImageNet ILSVRC 2012 images (as was used in previous work for clustering). We use cosine similarity between pairs of images and average linkage. In hierarchical clustering, the internal nodes typically correspond to clusters of their descendant leaf data points. We therefore use the transitive closure of the parent/child edges in the tree structure.

**K-Nearest Neighbor Graph**. Using the same the same ImageNet features as in the hierarchical clustering graph, we build a K nearest neighbor graph (using K=5). We use directed edges from $i$ to $j$ if $j$ is one of the K nearest neighbors of $i$. While these edges are the reverse of the nearest neighbor relationship, they better represent the asymmetric, containment properties that methods express via their conditional probability function. The edge distribution and structure of nearest neighbor graphs has been studied and often seen to exhibit preferential attachment characteristics [50, 43].

# I  Correlation between Graph Characteristics and Performances

In this section we extend the analysis in Section 6.2 to relative F1 among SIM, BOX, HYPERBOLIC embeddings and calculate Spearman correlation $\rho$ between relative F1 and graph characteristics such as average degree, clustering coefficient, flow hierarchy and reciprocity. Results are shown in Table 3 and Figure 5. From results, both Hyperbolic and box embeddings tend to prefer graphs with lower degree, lower clustering coefficient, higher flow hierarchy and lower reciprocity, indicating these geometric embeddings can better model directed graphs with less cycles and lower density. Comparing with HYPERBOLIC, BOX shows better tolerance against higher average degree and cluster coefficient, while it shows stronger preference over high flow hierarchy and low reciprocity.

| Relative F1 | Characteristic | $\rho$ |
|---|---|---|
| $F1_{BOX} - F1_{SIM}$ | Avg. Degree | -0.577 |
| | Cluster. Coefficient | -0.395 |
| | Flow Hierarchy | 0.402 |
| | Reciprocity | -0.336 |
| $F1_{HYPERBOLIC} - F1_{SIM}$ | Avg. Degree | -0.494 |
| | Cluster. Coefficient | -0.578 |
| | Flow Hierarchy | 0.204 |
| | Reciprocity | -0.162 |
| $F1_{BOX} - F1_{HYPERBOLIC}$ | Avg. Degree | 0.211 |
| | Cluster. Coefficient | 0.537 |
| | Flow Hierarchy | 0.121 |
| | Reciprocity | -0.113 |

Table 3: Spearman Correlation between Relative F1 and Graph Characteristics. All methods use 8 parameters per node.

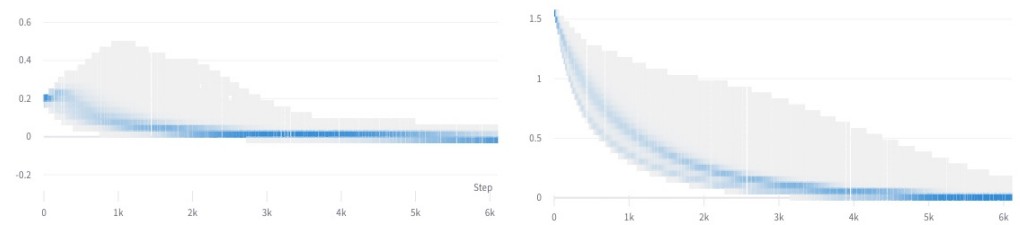

Figure 6: **Temperatures over time**. The left shows volume temperatures and the right shows intersection temperatures for a particular T-BOX model.

## J   Learned Temperatures

With the loose interpretation of temperatures as uncertainty, one might expect that models would benefit from higher temperatures (i.e. more uncertainty) at the start of training, and that these temperatures should decrease (i.e. less uncertainty) over time. This would allow easier training at the start, when boxes are initialized randomly and thus may need significant movement, while also affording greater representational capacity at the end, where boxes may actually need to be disjoint in order to model the sparsity of the adjacency matrix. In fig. 6 we see that this effect can be observed empirically, even without explicit regularization encouraging this behavior.