# OpenReview forum: "Capacity and Bias of Learned Geometric Embeddings for Directed Graphs"
_NeurIPS.cc/2021/Conference — NeurIPS 2021 Poster_

### Official Review · Reviewer_TVmD · 2021-07-07

**Rating:** 6
**Confidence:** 3

**Summary:**

The authors study geometric embeddings for directed graphs and propose a continuous relaxation of probabilistic box embeddings. They also analyze geometric embedding methods for modeling directed graphs by theoretical analysis and experiments.

**Limitations And Societal Impact:**

Yes. The authors addressed the limitations and potential negative societal impact of their work in the checklist.

**Main Review:**

Update After Rebuttal
---
I have carefully read the authors' responses. The explanation of why the trainable temperatures can help box embeddings avoid performance saturation in higher dimensions is reasonable. However, I still feel that there is a gap between the theoretical analysis and empirical implementations. In the further response, the authors provide a graph depicting the value of temperatures across all boxes as they are observed during training. It seems to me that many of the temperatures are far from zero, which violates the assumption in Theorem 2. In all, I recommend a weak accept for this paper.

---
Originality
---
The proposed box embedding model is a simple extension of that in [1], while the model in this paper allows learnable temperature parameters in the approximation of max operators. However, learning adaptive temperature parameters is not a new idea and it has been used in other scenarios, e.g., the computation of attention [2]. The authors may want to cite these papers and discuss the relationship between them and this work.

[1] Shib Sankar Dasgupta, Michael Boratko, Dongxu Zhang, Luke Vilnis, Xiang Lorraine Li, and Andrew McCallum. Improving local identiﬁability in probabilistic box embeddings. NeurIPS, 2020.

[2] Junyang Lin, Xu Sun, Xuancheng Ren, Muyu Li, and Qi Su. Learning When to Concentrate or Divert Attention: Self-Adaptive Attention Temperature for Neural Machine Translation. EMNLP 2018.


Quality
---
My major concerns about this paper are as follows.
1. Why the proposed model can outperform existing box embedding models is unclear. The authors claim that their method avoids performance saturation common to other geometric models in high dimensions, but why?
2. The authors claim that the learned temperatures capture uncertainty related to min and max coordinates for the box representing nodes. However, they did not clearly describe the exact meaning of uncertainty. Moreover, the authors may want to provide theoretical or experimental analysis to support the aforementioned claim.
2. It seems that the theoretical analysis in Section 4 has no strong connection with the proposed model. Does the analysis provide any guidance in model designing, or provide theoretical explanations for the effectiveness of the model?

Clarity
---
Overall, this paper is well-written. However, some important details are missing, which may confuse the readers.
1. What do capacity and bias mean? Though these two items appear frequently in the title, abstract, and introduction, the authors did not provide formal definitions for them.
2. The authors may want to include all the related work in the main text instead of the appendix.
3. I suggest the authors detail the objectives in Section 5, since it is critical for reproducing the results.

Significance
---
This work outperforms the state-of-the-art geometric embedding models in some experimental settings. The authors perform standard theoretical and empirical analyses.

Other Comments
---
1. Box embeddings are also applicable in tasks involved with multi-relational graphs, e.g., knowledge base completion [3], which is also mentioned in the abstract of this paper. The authors should discuss these works and conduct experiments on multi-relational data, if possible.

[3] Ralph Abboud, İsmail İlkan Ceylan, Thomas Lukasiewicz, Tommaso Salvatori. BoxE: A Box Embedding Model for Knowledge Base Completion. NeurIPS 2020.

**Time Spent Reviewing:**

6h

---

> ### Author Response · Authors · 2021-08-10
> **t-Box model is a continuous relaxation of the model analyzed theoretically, equivalent in the limit of zero temperature. Trainable temperatures allow for a better priory, starting with high uncertainty -> low as boxes configure themselves in space.**
>
> Thank you for your review! We respond to each section below.
>
> ## Originality
> The first paper you mention (your [1]) is cited in our work as [14]. We refer to this approach as "Box", which is described in lines 152-156 and contrasted with our "t-Box" model in section 3.1. We acknowledge our work is far from the first to generally consider training a temperature parameter, and will include a mention of [2] in our related work, however the application to training temperatures in this setting (i.e. graph representation, as well as box models and region-based representations more generally) is novel. Furthermore, our paper’s research hypothesis - investigating the representational capacity and bias of these models - has not been explored rigorously in the past, and the contribution of theoretical results proving that boxes can represent any DAG are new not only to the machine learning community but also to graph theory, where the concept of boxicity originated.
>
> ## Quality
>
> 1. Previous improvements to box embeddings have relied on smoothing the loss function to improve training (see our [25], or [14] as you mentioned). As we prove in our work, even the original box embedding model introduced in [55] has the capability to represent any graph, however it is not straightforward to train it, and this difficulty compounds in higher dimensions. On the other hand, while the smoothness introduced in [25] and [14] mitigate training difficulties, they also have an impact on the representational capacity of the models (eg. all boxes have nontrivial intersection). Empirically, we observe that temperatures initialized relatively high at the start of training remain high for some time while the positions of the boxes are optimized, and then gradually decrease toward zero as training increases, allowing our model to have the best of both worlds. We will include additional visualizations of this temperature training over time in the appendix.
> 2. In this setting, the underlying graph is known, i.e. there is no uncertainty to be learned, and thus the main benefit provided by the uncertainty is as a more reasonable prior at the start of training. Indeed, as described above, even without additional regularization the temperatures tend to go toward zero as training progresses, and the boxes are more certain of their location. This additional analysis will be included in the appendix.
> 3. The proposed model is a continuous relaxation of the model analyzed theoretically in Section 4, which is a limiting case of the temperature parameters going to zero. As a result of this theoretical analysis, we know that t-Box can not only approximate any given graph arbitrarily well, but furthermore the fact that edges are decided based on thresholding energy values implies that there exists some nonzero temperatures for which t-Box represents the given graph. The same argument also implies that the models of [25] or [14] have some setting of the temperature parameter for which the model will capture any graph exactly, however as one might expect the optimal temperature for representation may not be optimal for training, as we observe experimentally when comparing to Box. We mention this connection briefly in lines 242-244, however we will make it even clearer in the camera-ready.
>
> ## Other Comments:
> We agree that the BoxE model is interesting, however it actually mixes vectors and boxes together in a way which capitalizes on and requires the multi-relational setting (i.e. nodes are vectors, and relations are represented using boxes). The task of multi-relational settings are deserving of their own exploration, as the goal is often to share information between different relations to infer the presence of new edges. From a capacity and bias perspective, however, as observed for vectors in [3], multi-relational data can simply be viewed as simultaneously modeling distinct graphs for each relation.

---

> > ### Comment · Reviewer_TVmD · 2021-08-18
> > **Reply to the response**
> >
> > Thanks for your response.
> >
> >
> >
> > - I agree that Theorem 2 presents an interesting conclusion for box embeddings, but I still feel that the connection between the theoretical analysis and experimental results is weak. The main theoretical results (Theorem 2) apply in the limit where the box temperatures -> 0. However, since the authors use trainable temperatures, there is no guarantee that the trained model has small enough temperatures. The authors claim that almost zero temperatures are frequently observed empirically during training, but I cannot find the corresponding results. Moreover, if the leaned temperatures are almost zero, why not just set them to be a small positive number at the beginning and fix them during training?
> > - I am not fully convinced of the explanation why the trainable temperatures can help box embeddings to avoid performance saturation in higher dimensions. Since the learned temperatures are not guaranteed to be close to zero, the representational capacity of the proposed model is also not guaranteed to be higher than those of [24] and [14].
> >
> > I hope the authors can provide further explanations. In the current state, I will stick to my score.

---

> > > ### Author Response · Authors · 2021-09-02
> > > **Additional graph depicting temperatures -> 0 and discussion of impacts of temperatures on higher dimensional training**
> > >
> > > Thank you for your reply!
> > >
> > > - As you suggest, previous models (namely, “Box”) had tuned the temperatures as hyperparameters. What we observe is an inherent tradeoff between ease of training at the beginning (higher temperatures) and representational capacity (where the “crisp edges” of boxes which result from lower temperatures is useful). Specifically, we observe empirically that using a fixed temperature throughout (essentially the “Box” baseline) does not perform as well, and posit (based on observing the temperatures over time) this is because temperatures which work well at the start of training are not necessarily the same as those which work well toward the end. Similarly, as we observe empirically in the paper, the rate and value for which these temperatures change may be different for different nodes in the graph and dimensions of the box representation.
> > >
> > > Here we see a graph depicting the value of temperatures across all boxes as they are observed during training:
> > >
> > > https://ibb.co/18N9xck
> > >
> > > This was not previously included in the paper, but we would be happy to include this observation as well as a quantitative evaluation of this phenomenon in the final version.
> > >
> > > - The volume of a box is a product over it’s dimensions, thus if two boxes are disjoint in just one dimension they are disjoint entirely. This facet of box embeddings makes it more difficult to train in higher dimensions, as there are more opportunities for disjointness and thus lower gradient signals. Larger temperatures amount to larger overlaps when calculating the intersection volume between two boxes, and as such ameliorate the issue with training which is particularly prevalent in high dimensions. As mentioned above, however, it would not be sufficient to simply use a fixed large temperature for much the same reason - it is impossible to make two boxes effectively “disjoint”. It is for this reason we feel that trainable temperatures are particularly beneficial in higher dimensions.

---

> > > > ### Comment · Reviewer_TVmD · 2021-09-02
> > > > **Thanks**
> > > >
> > > > Thanks for your detailed reply. It has addressed most of my concerns. I strongly encourage you to include these discussions in the final draft. Now I lean towards supporting the acceptance of this paper and have raised my score to 6.

---

### Official Review · Reviewer_E4N8 · 2021-07-13

**Rating:** 7
**Confidence:** 4

**Summary:**

This paper analyses models for node embeddings of directed acyclic graphs (DAGs). It has 3 contributions:

i) it proposes a relatively straightforward extension of probabilistic box embeddings P-BOX [56], called t-BOX which is similar to the previous GumbleBox model [14], but has per-dimension and per-node intersection and volume temperature parameters (learnable) as opposed to just global parameters (as in GumbleBox). This extension is natural and the motivation is clear, but comes with the cost of doubling the number of parameters of the P-Box and GumbleBox models.

ii) theoretically, it extends the prior theory of [41] and [13] (which was analysing only undirected graphs) to work with DAGs and prove  that their t-Box model can represent any DAG (theorem 2, proof is an extension of [41]), giving an upper bound on number of dimensions and bits per box (proposition 3, proof is an extension of the result in [13]). Moreover, corollaries 1 and 2  are also the authors' contributions, being simple extensions of the result  in [56]. These theoretical results are important for understanding the representational capacity of existing DAG embedding models and I find this section nice and well written.

iii) Empirically, the authors evaluate the representational capacity (not generalization, just the "overfitting" ability) of several popular DAG embedding models on a variety of synthetic and real datasets. The choice of datasets is diverse and the empirical evaluation (e.g. removing the hyperparameters as confounding factors) is compelling, however, there are a few downsides that I will discuss below.

**Limitations And Societal Impact:**

yes, they did

**Main Review:**

======= Update after rebuttal ====

I am increasing my score to 7. See discussion.

=============================


First, I will touch the main required parts of this section, but my detailed comments will be given afterwards.

1) Originality:  I largely discussed this in the previous section. The t-BOX method is a straightforward extension of P-BOX. In its own is an incremental contribution, but given the other contributions of the paper, it is fair. Related work is missing important citations which would also be needed in the evaluation section (see below).

2) Quality: largely discussed in the previous section.

3) Clarity: paper is well written, with nice and detailed descriptions of methods, theory, experimental setups, etc.

4) Significance: largely discussed in the previous section, but there are important points that I will add below.


Important drawbacks of this work:

- empirical results (figure 1) do not show a clear advantage of the t-BOX models over previous BOX or other models. In the best case, it is comparable, but not clearly better. Fig 2 is showing more clear gains, but that is of course at the cost of doubling the number of parameters.

- important prior work is missing both to be cited and to be compared empirically against: for example, it is mentioned that hyperbolic models struggle with transitivity and use an heuristic choice of the scoring function (eq 6). These issues were addressed by the work of hyperbolic entailment cones ([1,2,3]) which also address capacity difficulties of order embeddings.

- looking at representational capacity (i.e. overfitting ability) is one facet of the problem, but for the community it is arguably equally or more important to understand the generalization capability of these models. Evaluating tasks like link-prediction was done in most of the prior work on DAG embeddings, and the lack of it in this paper is a weakness. This is especially important since the proposed t-BOX model does not seem to show clear cases where it clearly outperforms the BOX or the other models.

- Many of the real datasets are relatively small (what are the stastistics of KNN and Imagenet-hac datasets?) and by "small" I mean that many of the models are already almost perfect with 32 parameters. As a consequence, it is quite hard to see how good these models really are in high dimensions as it is claimed in the abstract of the paper. Why not use larger datasets where the effect could be more visible, such as the full WordNet?

All in all, I would be happy to increase my score if: i) the authors would include at least one generalization experiment on a real dataset, ii) the authors would compare against hyperbolic entailment cones. Also, I would be glad to see at least one large real dataset where all the baseline models (excl. t-BOX but inc. P-BOX) are far from perfect in the 32 parameters case.


[1] Hyperbolic Entailment Cones for Learning Hierarchical Embeddings, Ganea et al, 20`18

[2] Inferring Concept Hierarchies from Text Corpora via Hyperbolic Embeddings, Le et al 2019

[3] Hierarchical Image Classification using Entailment Cone Embeddings, Dhall et al, 2020


**Time Spent Reviewing:**

6

---

> ### Author Response · Authors · 2021-08-10
> **Clarification regarding t-Box performance, and an extended discussion related to the reasons we intentionally focus on evaluating representation as opposed to generalization**
>
> Thank you for the comprehensive review! Below are our responses to the list of drawbacks:
>
>
> 1. **Performance improvements:** In figure 1, we compare the performance of both vector-based and geometry-based models using 8, 32, and 128 parameters per node. We observe that:
>     1. Using 32 or 128 parameters per node, t-Box always shows better performance than other geometry embeddings, and shows competitive results with vector-based models.
>     2. Using 8 parameters per node, t-Box outperforms vector based models, and although it does not outperform other geometric embeddings we pointed out (in line 329) this is an extreme case, effectively using only 2-dimensional boxes. The boxicity of these graphs is almost certainly larger than 2, however we included these results so as to push these methods to their breaking point as well as observe which methods are able to more effectively represent graphs in this extremely low parameter-per-node setting (eg. Hyperbolic).
> 2. **Hyperbolic Entailment Cones:** Thank you! We will add these citations and make an effort to include the hyperbolic entailment cone baseline to the camera ready for all datasets. As we note below, box embeddings have previously been compared to hyperbolic entailment cones in [0].
> 3. **Generalization:** You are correct, in that our work includes a rigorous evaluation of the ability for models to represent a given graph as opposed to assessing their ability to generalize. This was done intentionally, as assessments of generalization depend entirely on the noise model.  Without a specified noise model, given a set of nodes and some subset of edges for training  any graph containing these edges is equally likely, and thus it is unclear how to evaluate models under this setting. Formally, let $G=(N,E)$ be the ground-truth graph, and $G' = (N, E')$ some subgraph chosen for training, with $E' \subseteq E$. If $S$ is the set of all graphs containing $G'$ is a subgraph, then without some noise model telling us how edges were selected for training we simply have a uniform prior over $S$. It is entirely unclear how to compare a model's performance in this setting - if the model ends up representing some graph $H$, even if $H=G$, then there exists some other graph $F \in S$ where $G'$ is once again valid training data, and we would not have accurately modeled $F$.
>
>     Specifying a noise model mitigates this to some extent, since it reults in a different prior over $S$, however knowledge of the noise model is often used to change the training procedure. For example, in Ganea et al. 2018 the training data always contained the transitive reduction, and were tested on the transitive closure, however the training process itself included knowledge of the transitive closure by omitting such edges from the set of negatives. If the noise model is known to remove edges from the transitive closure only, symbolically taking the transitive closure of the training data is arguably as reasonable as removing the transitive closure from the set of negatives during training, and this leads to perfect performance. Determining the “right” noise model for assessing real-world link-prediction is a very important but difficult question to address, however it is outside the scope of this work.
>
>     By focusing on reconstruction, where the problem is mathematically well-defined, we can still infer results about a model’s generalization capabilities. For example, if a model cannot represent a transitively closed graph when presented with the entire graph as training data then it will also struggle to generalize to it when presented with only the transitive reduction. On the other hand, if a model has a propensity for representing a certain class of graphs one would expect generalizations to be of similar form. Precisely determining the distribution over graphs which contain the training data as a subgraph is intractable, however by observing how well different models can represent a variety of graph types we can develop some level of understanding of their generalization abilities, as we discuss in section 6.1 and appendix J.
>
>     In short:
>     1. Generalization experiments are mathematically ill-posed without a specified noise model.
>     2. With a known noise model, existing implementation often adapt their training methods (eg. negative sampling) which are conflating factors we wanted to avoid.
>     3. Defining an interesting noise model which actually mimics the desired link-prediction for real-world tasks is, itself, a challenging unsolved problem
>     4. Representation is a mathematically well-defined problem which can be rigorously evaluated, while also providing valuable insights about a model’s generalization ability.
>
>     If you have a specific noise model which you feel would add significant value to the paper we would could certainly include generalization experiments using this model to generate train/dev/test datasets, however as observed above it is common to adapt the training method based on the noise as well. This seems a bit dubious, and it was our intent to avoid introducing these potentially conflating factors, as the mathematically well-defined question of graph representation already allows for reasonable conclusions related to generalization to be drawn. We would also add that box embeddings have already been compared to Hyperbolic entailment cones on the transitive closure of WordNet task from Ganea et al. 2018 (see [0], Table 2).
>
> 4. **Larger Graphs:** The dataset sizes were chosen to allow training on the full adjacency matrix (not just the set of edges). (Statistics for real-world datasets are provided in lines 285-289, in particular there are 8,192 nodes in the Imagenet-HAC and KNN graphs.) Training on the full adjacency matrix was desired since otherwise this task once again turns into a sort of generalization task (i.e. your training data can be viewed as a set of “known” cells of the adjacency matrix which were observed during training, and your task is matrix completion). As mentioned above, the choice of how to sample negative edges during training depends on some assumed properties of the graph (as mentioned above, in Ganea et al. (2018) the negative samples are randomly corrupted pairs which are not edges in the full transitive closure, and in general choosing the number of negatives implies some perceived bias about the graph sparsity). It was our intent to avoid introducing even more potentially conflating variables.
>
>     We believe the sizes are large enough to justifiably extend the conclusions to larger datasets, as the determining factor is often some fundamental property of the graph (eg. transitivity) which does not change in difficulty as the number of nodes change, and is not worth introducing potentially conflating variables related to the form of negative sampling required to train on larger datasets. If you still feel that additional experiments of this form would add value, however, we would be happy to include experiments on all of WordNet using some fixed form of negative sampling (eg. corrupting a positive) across all models.
>
> [0] Patel, Dhruvesh, and Shib Sankar. "Representing joint hierarchies with box embeddings." Automated Knowledge Base Construction (2020).

---

> > ### Comment · Reviewer_E4N8 · 2021-08-12
> > **reply**
> >
> > 1. Performance improvements:
> > - Results for the 4 real datasets using 32 or 128 parameters per node show that Sim, Bilinear and Complex achieve near perfect scores. Why would one use more complicated models in this case ? It would be compelling to include a real dataset where these models are not close to perfect for 32 or 128 parameters, and where T-Box is clearly better. For example, why not use the full WordNet ?
> > - I am willing to increase my score if the authors would present such an evaluation for a real dataset during the discussion period.
> >
> > 2. Hyperbolic entailment cones (HEC):
> > - HEC address both the problems of the capacity of order embeddings (OE), as well as the heuristic Poincar\’e embedding (PE) approach of embedding DAGs via eq 6. In my view, if one decides to use OE and PE as baselines, then one should also evaluate HEC to make a compelling argument that supports the superior behavior of box embeddings.
> > - As said, I am willing to increase my score if the authors would present an evaluation for HECs during the discussion period.
> >
> > 3. Generalization:
> > - Thanks for the extensive discussion.
> > - Prior work on Poincare Embeddings [35, 36] as well as knowledge graph embeddings (e.g. [33,34] among many others) do evaluate generalization for link prediction. ML pipelines are about generalization, otherwise I fail to see why they would be useful (exceptions exist, such as compression tasks).
> > - Re “given a set of nodes and some subset of edges for training any graph containing these edges is equally likely” → yes, but only if one assumes an independent Bernoulli distribution over each edge. In practice, assuming that shared patterns dictate the edge generation process is a sensible approach for ML models. Further, a reasonably weak explicit noise model might be sufficient for this task. As a concrete model example, one can predict if there is an edge (u,v) by concatenating the two node embeddings and applying a neural network on top to give a probability distribution (or use a different scoring model such as Bilinear, Sim, etc). To generate training/test data, given a large graph, one can assign some pairs of nodes as test data, and the rest as training data, without explicitly assuming any noise model.
> > - The subgraph assumption, $E’ \subseteq E$, is not necessarily the best assumption in my opinion. For example, one could argue that a more realistic noise model changes with a low probability edges into non-edges and vice-versa, being adapted to always generate DAGs (e.g. assuming a predefined node topological order and all edges complying with it).
> > - Regarding reconstruction giving some hints for generalization. I somewhat agree that if a model cannot overfit well, then it has less hope to generalize well. But still, the two tasks are distinct and there is little indication of how well generalization works given a great/perfect reconstruction.

---

> > > ### Author Response · Authors · 2021-09-02
> > > **HEC Evaluations and Further Experiments**
> > >
> > > Thank you for your reply! We address your points in more detail below, however the *tl;dr* is:
> > > - Over the past weeks we have implemented a version of Hyperbolic Entailment Cones in our evaluation framework and ran the same extensive tuning process on them so as to provide them a fair comparison with other models. We find that they are, indeed, quite competitive, and appreciate the suggestion to include them as it makes the paper more complete, particularly as they sit at the intersection of a vector and region-based model.
> > > - The main reason we evaluate on smaller datasets is that we train and evaluate over the entire adjacency matrix, thus removing any conflating factors in sampling and increasing our confidence that the evaluation is an accurate assessment of the model.
> > > - We absolutely agree that reconstruction and generalization are two separate tasks, and models which perfectly reconstruct may not generalize well. We also agree that the subgraph approach to analyzing generalization (which is quite standard in the literature) is not the best assumption, however defining a new "correct" form of generalization evaluation, particularly in the abstract setting of a graph without labels on the edges or nodes, deserves significant consideration in it's own right, and was outside the scope of this work.
> > >
> > > Even more detailed comments follow.
> > >
> > > 1. Performance Improvements
> > > - The aspect of interest to us regarding the relative performance of the models in 8, 32, and 128 parameters was that all other geometric models either saturated or even decreased their performance in higher dimensions, whereas the t-Box model does not. In addition, the vector models which are able to achieve nearly perfect scores in 32 and 128 parameters struggle to do so with only 8 parameters per node, while t-Box is quite competitive here. We find t-Box to be novel in this regard - better than vectors in low dimensions, and better than other geometric approaches in higher dimensions.
> > > - We agree that it would be desirable to evaluate these models on larger graphs where the vector models are not close to perfect with 32 or 128 dimensions, however in such a setting it is no longer possible to efficiently train on the full adjacency matrix, in which case additional confounding variables (eg. the form of negative sampling) begin to play a role. In addition, even simply evaluating these models on the full adjacency matrix is computationally prohibitive - the adjacency matrix for WordNet has ≈ 6.7 billion elements, taking up approximately 27GB if using float32 scores. Even sorting this in order to calculate the optimal threshold for F1 calculation is computationally intensive. While previous experiments have approximated this evaluation by using a test set with fixed random negatives (eg. 1:10, as done in Hyperbolic Entailment Cones), our own prior experiments have suggested such evaluations are not reliable. For example, experiments on the Mammal subset of WordNet demonstrated that a model which achieves over 0.9 F1 on a fixed test set may actually achieve less than 0.1 F1 when evaluated on the full adjacency matrix, thus indicating that it has not, in fact, learned the desired graph. We do agree it is worth assessing situations where the vector models fail, however, and will continue to work on the engineering problems necessary to train and evaluate reliably at these larger scales.
> > >
> > > 2. Hyperbolic Entailment Cones (HEC):
> > > We agree that HEC is a reasonable baseline to include as well. We implemented HEC in our evaluation framework, validated it’s correctness by reproducing the reported results from the paper, and ran our full evaluation suite on this model so as to include it as an additional baseline for in the representational capacity experiments. We will include these results in the final version of our work. Please see the full set of graphs here:
> > >
> > >
> > > https://ibb.co/KmWD70X
> > >
> > > Tabular results (as compared with OE, Hyperbolic, HEC, Box, and t-Box) are shown below:
> > >
> > > | ACC (8 params / node) |OE|Hyperbolic|HEC|Box|t-Box (/n/d)|
> > > |---|---|---|---|---|---|
> > > |Kronecker|0.331|**0.423**|**0.456\***|0.416|0.344|
> > > |Scale Free|0.248|0.511|**0.632\***|**0.549**|0.283|
> > > |Balanced Tree|0.594|**0.905\***|**0.827**|0.788|0.684|
> > > |nCRP|0.379|**0.891\***|**0.854**|0.738|0.495|
> > > |Price|0.695|0.759|**0.798**|**0.806\***|0.766|
> > > |Balanced Tree (TC)|0.850|0.832|**0.950**|**0.998\***|0.873|
> > > |nCRP (TC)|0.686|0.569|**0.945**|**0.981\***|0.811|
> > > |Price (TC)|0.824|0.832|0.918|**0.962\***|**0.940**|
> > > |Knn Graph|0.662|**0.704**|0.541|**0.712\***|0.619|
> > > |Imagenet-hac|0.615|**0.680\***|**0.638**|0.579|0.529|
> > > |Wordnet-animal|0.643|**0.915\***|0.823|**0.835**|0.746|
> > > |Wordnet-animal (TC)|0.949|0.966|**0.967**|**0.997\***|0.954|
> > >
> > >
> > > | ACC (32 params / node) |OE|Hyperbolic|HEC|Box|t-Box (/n/d)|
> > > |---|---|---|---|---|---|
> > > |Kronecker|**0.675**|0.606|0.504|0.616|**0.724\***|
> > > |Scale Free|**0.815**|0.700|0.746|0.810|**0.868\***|
> > > |Balanced Tree|0.910|**0.952**|0.836|0.845|**0.979\***|
> > > |nCRP|0.727|**0.924**|0.894|0.892|**0.962\***|
> > > |Price|0.835|0.803|0.786|**0.948**|**0.950\***|
> > > |Balanced Tree (TC)|0.998|0.978|0.982|**1.0\***|**0.998**|
> > > |nCRP (TC)|**0.998**|0.921|0.983|**0.999\***|0.998|
> > > |Price (TC)|0.988|0.884|0.943|**0.999\***|**0.996**|
> > > |Knn Graph|**0.858\***|**0.803**|0.535|0.789|0.798|
> > > |Imagenet-hac|**0.951**|0.694|0.670|0.682|**0.990\***|
> > > |Wordnet-animal|0.911|0.917|0.844|**0.942**|**0.993\***|
> > > |Wordnet-animal (TC)|0.997|0.975|0.975|**0.999\***|**0.999**|
> > >
> > >
> > > | ACC (128 params / node) |OE|Hyperbolic|HEC|Box|t-Box (/n/d)|
> > > |---|---|---|---|---|---|
> > > |Kronecker|**0.797**|0.666|0.554|0.684|**0.940\***|
> > > |Scale Free|0.807|0.703|0.785|**0.855**|**0.986\***|
> > > |Balanced Tree|0.906|**0.959**|0.839|0.880|**0.999\***|
> > > |nCRP|0.707|**0.908**|0.899|0.898|**0.993\***|
> > > |Price|0.891|0.836|0.847|**0.975**|**0.993\***|
> > > |Balanced Tree (TC)|0.999|0.987|0.988|**1.0\***|**1.0\***|
> > > |nCRP (TC)|0.999|0.891|0.989|**1.0\***|**0.999**|
> > > |Price (TC)|0.992|0.923|0.948|**0.999**|**0.999\***|
> > > |Knn Graph|**0.910**|0.808|0.553|0.787|**0.975\***|
> > > |Imagenet-hac|**0.898**|0.724|0.688|0.785|**0.997\***|
> > > |Wordnet-animal|**0.935**|0.910|0.837|0.859|**1.0\***|
> > > |Wordnet-animal (TC)|0.999|0.968|0.977|**0.999**|**1.0\***|
> > >
> > > Bold w. star: Best performance
> > > Bold w/o star: Second best
> > >
> > > Overall, we find that HEC are indeed a strong baseline, sharing some of the properties of hyperbolic space as well as the region-based models. As expected, they perform particularly well on transitively closed graphs, however like other geometric models they seem to saturate their performance with regard to dimension.
> > >
> > > 3. Generalization: We agree that the task of generalization is also important, and that the reconstruction and generalization are distinct - a model capable of perfect reconstruction may not generalize well, indeed the training process for most generalization tasks is equivalent to the same training as would have been performed in order to represent the edges in the training set only, and thus a model capable of perfect reconstruction would represent exactly those edges observed during training and no more. We also agree that the subgraph assumption (which has most predominantly been used in prior work) is not the best when assessing generalization, however as you point out there are many reasonable choices for a noise model. Different noise models may result in a different choice of model and even training (eg. the style of negative sampling during training), and assessing all such combinations in the abstract was not possible in the scope of this work. Nevertheless, we believe our theoretical and empirical results on representational power are useful on their own merits, and are consistent in scope with previous works on graph embeddings [1,2]. We hope that our work has provided a comprehensive survey of existing graph embedding techniques and a rigorous evaluation of their representative capacity, which practitioners attempting to model graphs in the context of a larger deep learning architecture (where generalization may result from other areas - eg. a pretrained language model) will find useful.
> > >
> > > [1] F. Sala, C. De Sa, A. Gu, and C. R ́e.“Representationtradeoffs for hyperbolic embeddings.”ICML, 2018.
> > >
> > > [2] R. Bhattacharjee, and S. Dasgupta.“What relations arereliably embeddable in Euclidean space?.”ALT, 2020

---

> > > > ### Comment · Reviewer_E4N8 · 2021-09-02
> > > > **good**
> > > >
> > > > Thanks for your efforts.
> > > >
> > > > HEC evaluation looks great.
> > > >
> > > > Re generalization: I agree that it requires an extensive analysis as a separate study.
> > > >
> > > > Re higher number of parameters: I agree on the computational difficulties. However, as a practitioner, looking at your paper makes me convinced that using simple Sim or Bilinear models with appropriate number of parameters (e.g. 128) is the best choice for both synthetic and real datasets. But all in all, I think your empirical evaluation is extensive and useful for future progress in this area.
> > > >
> > > > I am increasing to 7.

---

> > > > > ### Author Response · Authors · 2021-09-02
> > > > > **Thank You**
> > > > >
> > > > > Thank you for reconsidering your evaluation and your helpful suggestions during this rebuttal period.

---

### Official Review · Reviewer_6Ekw · 2021-07-15

**Rating:** 7
**Confidence:** 3

**Summary:**

# A solid work on Geometric Embeddings

This paper discusses the properties of recent geometric embeddings for representing graphs and extends a new continuous relaxation of the probabilistic box embeddings. This paper is written is a clear way: the motivation and contribution are clear, and backed up by extensive experiments. Theoretical contributions, especially the precision bound for box embeddings, are presented in parallel with its methodological contribution. Based on these strengths, although there are some issues (discussed below), I give a clear accept of this paper.


**Limitations And Societal Impact:**

The underlying mechanism why axis-wise temperature helps in the high-dimensional case is not studied. I would recommend the authors to at least give an intuitive explanation.

**Main Review:**

This paper discusses the properties of recent geometric embeddings for representing graphs and extends a new continuous relaxation of the probabilistic box embeddings. I think this is a solid work what worth a clear acceptance. Yet I do want to point out certain issues and suggest further improvements:
* Discussions on hyperbolic embeddings are just scratching the surface. Theoretical results are centered on box embeddings. I would suggest the authors to either give a more detailed discussion on hyperbolic embeddings, or shorten them.
* The mechanism why axis-wise temperature gives better performance in high-dimensional scenarios is not discussed at all, which would be important to understand the source of performance gain (although all models are tuned with Bayesian hyperparameter search).
* Given this paper is more about box embeddings, I would suggest to shorten discussions on complex and hyperbolic vectors (put them in the appendix) and make room for the experiments. In the experiments, more studies about why axis-wise temperature improves performance in high-dimensional scenarios should be conducted (Possibly to show more stable optimization / numerics, as these are the major challenges for the hyperbolic case).

Other Minor Points:

- For important theoretical results, their proofs should be sketched briefly (e.g.,  Thm 2)
- I would encourage using simple language to explain what the theorem says intuitively. For example, for proposition 3, the authors may say box embeddings can represent graphs with exponentially large number of nodes (because of the log V term), but the connections should not be too dense (because of the linear $\Delta$ term)
- The math behind why POE always yield a DAG but OE does not should be briefly explained, my understanding is that the composition of $max(\cdot)$ and L1 norm allows separation of variables while L2 norm does not
- Important concepts in the theorems should be explained briefly (e.g., what does bits per box mean?)

**Time Spent Reviewing:**

8

---

> ### Author Response · Authors · 2021-08-10
> **Thank you for the suggestions, and a further explanation of intuition behind performance increase due to axis-wise temperatures**
>
> Thank you for the detailed review and suggestions! We address your specific questions below:
>
> **Hyperbolic / Complex Baselines:** You are correct that the focus of this paper is on box embeddings and not on the complex or hyperbolic baselines, which have similarly been explored theoretically [3, 43]. This work is the first to engage in such a large-scale experimental evaluation of all these methods, however, and thus our goal was to make it self-contained and suitable for use as a survey of current approaches. In addition, hyperbolic embeddings do not actually have a standard approach for training on directed graphs. We attempted to provide the minimal background necessary to understand our definition of the energy function (9) which is a combination of existing ideas [35, 23], however we will attempt to further compress this section and perhaps move some of the motivation (lines 104-119) to Appendix A.
>
> **Axis-Wise Temperatures**: An intuitive explanation for why axis-wise temperatures improve performance (apart from simply having more trainable parameters) is that a $d$-dimensional box model acts as a conjunction of $d$ 1-dimensional box models, in which some are more “certain” than others. Thus, the dimensions which are in highest agreement with the data can become more “certain” (i.e. use lower temperature) quickly, while the others preserve a higher temperature, allowing for easier training. As explored in the additional ablation experiment (see Figure 2, and lines 326-333) our conclusion is not that axis-wise temperatures are particularly good in higher dimensions, but rather that comparing models with an equal number of parameters per node means that, for 8 parameters per node, the t-Box (/n/d) model is required to use 2-dimensional boxes. The additional training flexibility provided by axis-wise temperatures is not enough to overcome the limitation of this extremely low-dimensional geometry - i.e. the graphs we seek to represent simply have boxicity larger than 2. Of course for any practical use it would be quite reasonable to use a dimension larger than 2, and even in the 8-dimensional case (parameters per node = 32) we see that the t-Box (/n/d) model generally shows better performance.
>
> **Minor Points:** We agree that having proof sketches and intuitive explanations of the more technical details of the paper would add value, some of which were removed in order to meet the page limit. We will attempt to restructure the paper to add these elements back.

---

> > ### Comment · Reviewer_6Ekw · 2021-08-17
> > **Thank you for your response**
> >
> > * Now I can see their motivation about including the hyperbolic embeddings. Although I think the motivation, i.e., viewing different models with different energy functions, is clear, it does not change my view that the contribution of this paper is not related to hyperbolic embeddings. -- Yet I think this has becomes a minor point given the current concerns
> > * However, I am not convinced by the current explanation why making temperature trainable could allow the certain more certain (this is wrong), there are three important points:
> >     * The "certainty", or the dimensions that logsumexp to a high value, can be easily constructed by setting the largest $x_i$ within $x$ to be larger than other $x_j, j \ne i$ in equation 15, without a small temperature. E.g., try LSE$([1, 10], 1) = 10.0001$ in pytorch. It is the variance var$(x / t)$, which comes from the interplay of var(x) and t, that jointly determines the certainly, and the model has its full right to "compensate" small t with larger var(x).
> >     * The training dynamics should also be considered. In the early stages of training, the model may choose to (a) exploitation: converges quickly to be more certain about existing solutions it finds, or (b) exploration: stays more uncertain and explores larger parameter space. This is a tradeoff between exploration and exploitation, and one cannot say exploitation is always better.
> >     * The gradient path with/ without tunable temperature, is different, which I think might be a good starting point. Instead of saying "become more certain quickly", the authors may consider "exploits the current parameter subspace"; instead of saying "easier training", the authors may consider "the gradient path is split to both x and t".
> > * Yet the above discussions, which already concern gradient structures and training dynamics, may be beyond the scope of this paper. So maybe as deep as space permits.
> > * About the saturated performance in the high-dimensional case, I fully agree with reviewer E4N8.
> >
> > Given the current discussions, I would like to stick with my current score.

---

> > > ### Author Response · Authors · 2021-09-02
> > > **Minor additional comment related to temperature training dynamics**
> > >
> > > Thank you again for your comments, we will attempt to incorporate your suggestions into the final version, in particular compressing the section on hyperbolic embeddings.
> > >
> > > A minor clarification related to your comments about temperatures:
> > > By "certainty", we were referring to the temperatures specifically, as their interpretation in the Box model was as the scale parameters for the latent Gumbel random variables which comprise the corners of a box. In that setting, the scale parameter acts as a sort of meta-probabilistic uncertainty regarding the box coordinates. You are correct to point out that overall box volume is a combination of both temperature and side-length, and a large box may be obtained with either large side-length and low temperature or small side-length and high temperature (or both). In our experiments, however, we typically observe the following pattern of temperatures: https://ibb.co/18N9xck . This naturally maps to the "exploration" (high temperatures) and "exploitation" (low temperatures) you mentioned. Here is another plot (https://ibb.co/47QVg36) which shows the mean temperatures for intersection and volume for a different run, where we can clearly observe the model's preference to explore more in the beginning and then, once the boxes are located in a reasonable configuration, the model begins to exploit. The interesting thing we find is that the model seems to effectively choose it's path along these lines without any explicit regularization. ​We will include a qualitative discussion as well as additional quantitative analysis of this phenomenon in the final version of the paper.
> > >
> > > We hope we have fully addressed all your comments and suggestions, and thank you again for your effort reviewing our work.

---

### Official Review · Reviewer_YYhw · 2021-07-18

**Rating:** 6
**Confidence:** 4

**Summary:**

This work provides an empirical and theoretical foundation for the analysis of geometric representations of directed graphs. The paper introduces a novel box embedding model, extending the recent Gumbel-box process to learn a parametric model representing a continuous relaxation of box intersections. The model can trade-off between smoother, vector-like settings and more discrete, region-like settings, and provides better performance across a range of dimensionalities.

Directed graphs have received considerably less attention in the representation learning community than undirected graphs, making this paper particularly appealing.

**Ethical Concerns:**

None.

**Limitations And Societal Impact:**

Limitations of the work are described. This work does not seem to introduce any new potential for negative societal impacts.

**Main Review:**

(1) Extensive and meaningful experiments with new observations can inform the design of future algorithms. Insightful empirical analyses are a strong plus of the proposed work. Include error bars in Figure 2.

(2) Description of the novel box embedding method in section 3 lacks some information. Please describe the complete algorithm using the algorithmic environment. What are the critical hyperparameters of the approach? How does the new approach scale with graph size? I couldn't find this information in the Appendix.

(3) The authors treat the prediction of a directed edge from node i to j as a binary classification problem (in section 5). How are reverse edges predicted? Are those treated independently? The appropriate edge scoring function depends on the type of geometric model (e.g., Nickel and Kiela's [35] scoring function was designed for hyperbolic vectors). Would you mind clarifying and providing details for each method included in the benchmarking study?

(4) The authors carry out a suite of experiments on synthetic and real-world graphs, comparing various vector, hyperbolic, cone, and box representations over a range of dimensions. That is great, but it also means that some aspects of the new model are not thoroughly evaluated (i.e., see point 2).

**Time Spent Reviewing:**

1

---

> ### Author Response · Authors · 2021-08-10
> **Clarification regarding error bars, hyperparameters, edge direction, and discussion related to scalability of results**
>
> Thank you for your detailed review and questions, which we address individually below:
>
>
> 1. **Error Bars:** Thank you for your laudatory comments! Regarding error bars - in Figure 1 these are calculated with respect to sampling graphs from various generative models (please refer to line 296-302), and thus capture the level of variation in performance when modeling different graphs from a given generative model. Figure 2 depicts real-world graphs, however, and thus there is no randomness on the level of the graph to calculate error bars with respect to.
> 2. **Algorithm and Model Clarifications:** Thank you for the suggestion, we would be happy to describe the complete algorithm using the algorithmic environment in the appendix. The critical hyperparameters are the same as for the other methods considered - embedding dimension and learning rate. The choice of temperature initialization and bounds in the t-Box model can be viewed as a hyperparameter, taking the place of the global (fixed) temperature hyperparameters from the Box model, however as these are now trainable t-Box is far less sensitive to their initialization. The computational complexity of the t-Box model is simply a constant factor difference when compared with Box, and thus will scale equivalently with respect to graph size and dimension.
> 3. **Edge Prediction:** We wholeheartedly agree that the method of predicting edge direction must be adapted to the geometry of the model. In our case, each model has an appropriate asymmetric energy function: in $E_{SIM}, E_{BILINEAR}, E_{COMPLEX}$ (eq. 1 - 5) the transformation matrix $W$ is asymmetric, and in $E_{OE}, E_{POE}$ (eq. 10, 11), coordinatewise-dominance (via the $\max$ operation) provides this asymmetry. For box models, we see that $E_{BOX}$ (eq. 13, 18) is clearly asymmetric; geometrically, the box which we take relative volume with respect to differs depending on the edge direction.  For $E_{HYPERBOLIC}$ (eq. 9) the second term indicates that edges should go from nodes with smaller Euclidean norm to larger. As we discuss in lines 104-119, the motivation is similar to that in Nickel and Kiela's [35], where the authors trained a symmetric distance between two points in Poincaré space and used Euclidean norm to decide the direction as described in eq. 6. In lines 113-115 (and Appendix A) we mention that this score function is quite sensitive, leading to our proposed eq. 9 that can be used for both training and test. In each case, we evaluate over the full adjacency matrix, where both directions are predicted separately (please refer to line 76-82).
> 4. **Additional Aspects:** We interpret the aspects of interest referred to here to be the influence of various hyperparameters and the extent to which performance scales with graph size. **Hyperparameters:** We used extensive hyperparameter tuning in order to ensure we were comparing the optimal performance of each model. Since the number of parameters per node has the most significant impact on performance (both as suggested by previous work as well as our observed results) we intentionally focused on this comparison in Figure 1, however our resulting experimental data (which we will make public) can also be used to infer relationships between other hyperparameters (eg. learning rate, batch size) and performance for each model. In particular, as mentioned in our response to (2), the only different hyperparameters of our new model compared to the previous Gumbel box approach is the choice of temperature initialization and bounds. As expected, our model is somewhat less sensitive to the initialization of temperature as compared to previous Gumbel box models, wherein the temperature was a (fixed, global) hyperparameter. **Graph Size:** As noted, our focus in the experimental section of this work was to rigorously evaluate the claims posited in previous work that different graph representations have inductive biases which make them more or less amenable to representing certain types of graphs (eg. trees, transitively closed graphs, etc.), and in particular evaluate the extent to which a model which can provably represent any graph (t-Box) can actually be trained to do so. The properties of these graphs are such that it is intuitively plausible for the models to maintain their relative performance as the number of nodes increase, however we believe to properly address this would require it’s own similar level of empirical evaluation, which we leave for future work.

---

> > ### Author Response · Authors · 2021-09-02
> > **Any additional questions or concerns?**
> >
> > As the discussion period nears it's end, we wanted to reach out once more and verify that we have sufficiently addressed your questions. In short:
> >
> > 1. Error bars are depicted with respect to different graphs sampled from the generative models in the synthetic case, and thus are not applicable to the collection of real-world graphs.
> > 2. We will happily include a description of the algorithm using the algorithmic environment in the appendix. In short, we are updating model parameters based on gradient descent, and the novelty of the proposed model comes from defining an energy function (eq. 18) with temperature parameters that are trained. The hyperparameters of our model are essentially the same as with the previous Box model.
> > 3. For each method we decide edge existence based on thresholding their energy function, where this threshold is chosen such that the F1 of predicting an edge is largest. The energy functions themselves are adapted for different geometric representations, in-line with prior work.
> > 4. We provided some additional discussion related to hyperparameters of our model in our response above, as well as in our response to reviewer TVmD, where we provided the following graph depicting the value of intersection and volume temperatures as the model trains:
> > https://ibb.co/18N9xck
> > This depicts the training pattern alluded to in the paper, wherein temperatures start relatively high (to provide easier training) but gradually decrease to zero (for better representational capacity). An interesting point is that this occurs without explicit regularization. We will include a qualitative discussion as well as additional quantitative analysis of this phenomenon in the final version of the paper.
> >
> > Please let us know if there are any remaining questions we can address!

---

### Decision · Program_Chairs · 2021-09-27

**Decision:**

Accept (Poster)

**Comment:**

This paper has reached a consensus in that the authors are offering several useful items to the study of embeddings, including a new method that balances between the flavors of two popular classes of existing embedding methods. This is a relatively practical area with many different proposed methods, so the authors approach—to bridge known gaps and provide useful theoretical and empirical insights—is welcome. The reviewers have very little variance between them; post clarifications by authors almost all the scores are identical. I also largely agree with the paper’s strengths. I’m positive about the contributions offered here.

Perhaps the main argument against acceptance, as also noted by several reviewers, is that the paper is a bit of a grab-bag in a sense, in that there’s some theoretical analysis of existing methods, the introduction of a new method, along with a large empirical study (not necessarily focused on just evaluating the proposed method). This is not a major problem, since all of these parts are useful, provide interesting insights that are likely to be helpful to practitioners, and the authors’ writing quality in weaving together these concepts is pretty good. The additional discussion period helped clarify a number of things here as well.

I think the authors made a wise choice to study the basics and paint a fairly complete picture (e.g., the focus on representation quality rather than generalization, attempting to control all the factors involved in quality in the empirical results). This is a more well-grounded and scientific approach than is typically found in papers in this area.